# Rethinking Residual Distribution in Locate-then-Edit Model Editing

**Xiaopeng Li**   **Shangwen Wang**   **Shasha Li**[*]   **Shezheng Song**   **Bin Ji**   **Jun Ma**[*]   **Jie Yu**[*]
National University of Defence Technology
{xiaopengli,wangshangwen13,shashali,ssz614,jibin,majun,yj}@nudt.edu.cn

## Abstract

Model editing enables targeted updates to the knowledge of large language models (LLMs) with minimal retraining. Among existing approaches, locate-then-edit methods constitute a prominent paradigm: they first identify critical layers, then compute residuals at the final critical layer based on the target edit, and finally apply least-squares-based multi-layer updates via **residual distribution**. While empirically effective, we identify a counterintuitive failure mode: residual distribution, a core mechanism in these methods, introduces weight shift errors that undermine editing precision. Through theoretical and empirical analysis, we show that such errors increase with the distribution distance, batch size, and edit sequence length, ultimately leading to inaccurate or suboptimal edits. To address this, we propose the **B**oundary **L**ayer **U**pdat**E (BLUE)** strategy to enhance locate-then-edit methods. Sequential batch editing experiments on three LLMs and two datasets demonstrate that BLUE not only delivers an average performance improvement of 35.59%, significantly advancing the state of the art in model editing, but also enhances the preservation of LLMs' general capabilities. Our code is available at https://github.com/xpq-tech/BLUE.

## 1   Introduction

Large language models (LLMs) possess powerful comprehension and generation capabilities and have become foundational infrastructure for various AI applications. However, the knowledge encoded in the parameters of LLMs is limited to the training data and cannot be updated to reflect changes in world knowledge. Updating the parameters of LLMs through retraining to keep them in sync with world knowledge entails high computational costs [1]. Recently, model editing has garnered increasing attention as a promising technique for efficiently updating the parameterized knowledge in LLMs, which aims to correct erroneous or outdated knowledge within LLMs without compromising their other capabilities [2].

Locate-then-edit methods are a prominent family of model editing techniques. They treat the Feed-Forward Network (FFN) as a key-value memory [3] and update the key layers responsible for storing factual knowledge using a least-squares solution. Specifically, these methods first employ causal tracing analysis to identify multiple critical layers within LLMs that encode factual information. They then use optimization techniques to compute the residuals required to update the final critical layer. Finally, the residuals are distributed evenly from the first to the last critical layer, and the updates are applied using a least-squares solution [4, 5], a process referred to as **residual distribution**.

Although locate-then-edit methods have achieved remarkable performance on model editing tasks [6], **we identify a counterintuitive failure mode: residual distribution, a core mechanism in these methods, introduces weight shift errors that undermine editing precision.** Specifically, we

---

[*] Corresponding Authors.

39th Conference on Neural Information Processing Systems (NeurIPS 2025).

first empirically demonstrate that the contribution of residual distribution to model editing diminishes as the distribution distance increases and that the distributed residual is not the optimal residual for editing. Subsequently, we theoretically find that the upper bound of weight update errors increases with: (a) the size of the editing batch, (b) the number of sequential edits, and (c) the residual distribution distance. These findings indicate that **residual distribution can actually negatively impact the model editing of the locate-then-edit approaches**.

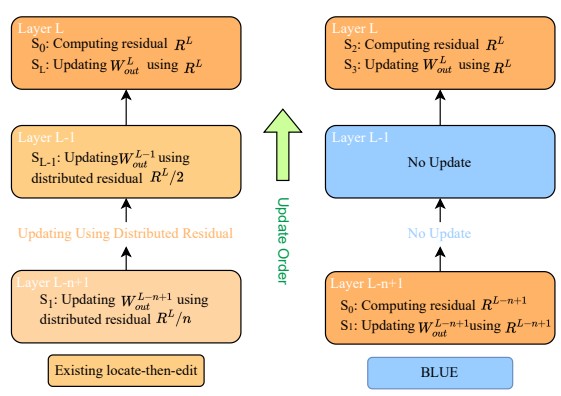

Figure 1: Comparison of existing locate-then-edit methods and BLUE. The $n$ critical layers are edited, where $L$ denotes the last critical layer.

Therefore, we propose the **B**oundary **L**ayer **U**pdat**E** (**BLUE**) strategy, which enhances locate-then-edit methods by updating only the first and last critical layers through direct computation of residuals, without residual distribution. The comparison between BLUE and existing locate-then-edit methods is shown in Figure 1. We apply BLUE to enhance MEMIT [5], RECT [7], PRUNE [8] and AlphaEdit [6]. Results from 12 sequential editing experiments conducted on three LLMs and two datasets show that BLUE improves the performance of existing locate-then-edit methods by an average of **35.59**%. Our further analysis on downstream tasks and representation shift demonstrates that BLUE also enhances the ability of locate-then-edit methods to preserve general capabilities and mitigates representation shifts in the post-edit LLMs. Furthermore, we show that BLUE not only improves editing efficiency, but also strengthens the performance of locate-then-edit methods in long-form model editing [9, 10] scenarios. In summary, our contributions are as follows:

- Through empirical and theoretical analysis of residual distribution, we reveal that residual distribution of locate-then-editing methods actually leads to inaccurate model editing.

- We propose the BLUE strategy, which discards residual distribution and enhances existing locate-then-edit methods by updating only the first and last critical layers through direct residual computation.

- Experimental results show that locate-then-edit methods enhanced with BLUE outperform the original methods, better preserve LLMs' general capabilities, and further mitigate representation shifts in the post-edit LLMs.

## 2 Related Work

Model editing can be categorized into **parameter-preserving** and **parameter-modifying** approaches, depending on whether the original model parameters are altered.

**Parameter-preserving** model editing employs techniques like prompt engineering or attaching additional parameters [11–14]. A representative method for prompt engineering is IKE [15], which retrieves $n$ contexts of the edited knowledge for a query to guide the model's response without altering its internal parameters. Methods that attach additional parameters include SERAC [16] and GRACE [17], which store new memories externally and internally, respectively, by introducing new parameter modules.

**Parameter-modifying** approaches achieve model editing by directly or indirectly adjusting model parameters [18, 10]. Direct methods, such as FT-L [19], perform constrained fine-tuning on a small number of layers to integrate new knowledge. Indirect methods can be divided into **meta-learning** and **locate-then-edit** methods. **Meta-learning** methods, like MEND [20], leverage a hypernetwork to transform edit-related representations and gradients into parameter updates. In contrast, **locate-then-edit** methods, such as ROME [4] and MEMIT [5], adopt a key-value memory perspective to identify and update single or multiple critical layers using least-squares optimization. Among these, locate-then-edit methods have gained popularity and inspired several variants, such as PMET [21], which focuses on precise editing, and AlphaEdit [6], which enhances the retention of original knowledge and strengthens sequential editing capability.

# 3 Background

## 3.1 Model Editing Problem

Model editing aims to efficiently update the knowledge of LLMs so that they remain in real-time sync with reality [22]. Factual knowledge changes rapidly, making its update in LLMs a pressing need. Factual knowledge can be represented as a triplet $(s, r, o)$, where $s$ is the subject, $r$ is the relation, and $o$ is the object. This knowledge can be transformed into a prompt $p_i + o$, where $p_i \in \mathcal{P}$ is an element of the set $\mathcal{P}$ that expresses the semantics of $(s, r)$ in natural language. The element that most directly expresses the semantics of $(s, r)$ is called $p$, while the rest elements are $p_r$. The goal of model editing is to redirect the object $o$ in the triplet to a new object $o^*$, represented as $t = (s, r, o) \rightarrow o^*$. To evaluate whether the post-edit model is effective on the post-edit knowledge triplet and does not affect other triplets, assessments are made from three aspects: efficacy, generalization, and specificity. Efficacy evaluates whether the model's prediction on $p$ is redirected to $o^*$. Generalization evaluates whether the model's prediction on $p_r$ is redirected to $o^*$. Specificity evaluates whether the model maintains its original predictions on inputs outside the set $\mathcal{P}$. To evaluate the generative capability of the post-edit model, [4] also uses fluency and consistency as evaluation metrics. Fluency measures the degree of repetition in the text generated by the model after editing; higher repetition indicates lower fluency. Consistency evaluates the degree of alignment between the content generated by the post-edit model based on $s$ and the reference text of the subject associated with the new object $o^*$. For more details, please refer to [4].

Model editing can be categorized according to whether the editing is sequential and the batch size into sequential editing, batch editing, and sequential batch editing [23]. Sequential editing refers to the continuous editing of a single piece of knowledge, while batch editing involves editing multiple pieces of knowledge at once. Sequential batch editing combines these two scenarios, involving the sequential editing of batch knowledge. This problem definition is highly relevant to the ever-changing nature of bulk knowledge in practice. Therefore, we directly present the problem of sequential batch editing. Suppose there is a sequence of $n$ knowledge sets to be updated: $[\mathbb{T}_1, \mathbb{T}_2, \ldots, \mathbb{T}_n]$, where each knowledge set $\mathbb{T}_i = \{t_1, t_2, \ldots\}$. Sequential batch editing requires that after performing $n$ sequential batch edits, the post-edit model can successfully predict all $n$ knowledge sets without affecting knowledge outside these sets.

## 3.2 Locate-then-Edit Model Editing

The locate-then-edit model editing is one of the most popular series of model editing methods [1]. These approaches typically use causal tracing [4] to identify the critical layers $\mathcal{L}$ where knowledge is stored, and then compute weight shifts using least squares modeling to update the weights.

Specifically, they view the feed-forward network (FFN) as key-value memories [3]. Let $\boldsymbol{h}^{l-1}$ be the residual stream of the $l-1$ layer, and $\boldsymbol{a}^l$ be the output of the self-attention block of the $l \in \mathcal{L}$ layer. The key-value memories of the FFN can be represented as follows:

$$\underbrace{\boldsymbol{m}^l}_{\text{value}} = \boldsymbol{W}_{\text{out}}^l \underbrace{\sigma(\boldsymbol{W}_{\text{in}}^l \gamma(\boldsymbol{h}^{l-1} + \boldsymbol{a}^l))}_{\text{key} := \boldsymbol{k}}, \tag{1}$$

where $\boldsymbol{m}^l$ is the output of the FFN block, and $\boldsymbol{W}_{\text{in}}^l$ and $\boldsymbol{W}_{\text{out}}^l$ are the input and output mapping weights of the FFN block, respectively. $\sigma$ and $\gamma$ are activation functions. $\boldsymbol{W}_{\text{out}}^l := \boldsymbol{W}_0^l$ is viewed as a linear associative memory that associates keys and values:

$$\boldsymbol{K}_0^l = [\boldsymbol{k}_1^l | \boldsymbol{k}_2^l | \ldots | \boldsymbol{k}_n^l], \boldsymbol{M}_0^l = [\boldsymbol{m}_{0,1}^l | \boldsymbol{m}_{0,2}^l | \ldots | \boldsymbol{m}_{0,n}^l]. \tag{2}$$

Before editing, the linear associative memory satisfies:

$$\boldsymbol{W}_0^l = \arg\min_{\boldsymbol{W}} \left\| \boldsymbol{W} \boldsymbol{K}_0^l - \boldsymbol{M}_0^l \right\|^2. \tag{3}$$

When new memories need to be inserted, a new group of keys $\boldsymbol{K}_1^l$ and values $\boldsymbol{M}_1^l$ will be updated into $\boldsymbol{W}_0^l$. Thus the new weight should satisfy:

$$\boldsymbol{W}_1^l = \arg\min_{\boldsymbol{W}} \underbrace{\left\| \boldsymbol{W} \boldsymbol{K}_0^l - \boldsymbol{M}_0^l \right\|^2}_{\text{preserve old}} + \underbrace{\left\| \boldsymbol{W} \boldsymbol{K}_1^l - \boldsymbol{M}_1^l \right\|^2}_{\text{insert new}}. \tag{4}$$

Let $\boldsymbol{W}_1^l = \boldsymbol{W}_0^l + \Delta^l$ where $\Delta^l$ is weight shifts. By applying the normal equation to Eq. (4), its closed-form solution can be written as:

$$\Delta^l = \boldsymbol{R}^l \boldsymbol{K}_1^{l\,T} \left( \boldsymbol{K}_0^l \boldsymbol{K}_0^{l\,T} + \boldsymbol{K}_1^l \boldsymbol{K}_1^{l\,T} \right)^{-1}, \tag{5}$$

where $\boldsymbol{R}^l = \left( \boldsymbol{M}_1^l - \boldsymbol{W}_0^l \boldsymbol{K}_1^l \right)$ is the residual of the new memories when evaluated on old weights $\boldsymbol{W}_0^l$ [5]. $\boldsymbol{K}_1^l$ and $\boldsymbol{K}_0^l$ are computed for each layer. In most locate-then-edit methods [5, 6], the residual of layer $l$ is evenly distributed from the residual of last critical layer $L = \max(\mathcal{L})$:

$$\boldsymbol{R}^l = \frac{\boldsymbol{R}^L}{L - l + 1} = \frac{\boldsymbol{M}_1^L - \boldsymbol{W}_0^L \boldsymbol{K}_1^L}{L - l + 1}, \tag{6}$$

where $\boldsymbol{M}_1^L = [\boldsymbol{m}_1^L | \boldsymbol{m}_2^L | ... | \boldsymbol{m}_u^L]$ represents $u$ entries of new memories. Each entry is computed using the following formula:

$$\boldsymbol{m}_i^L = \boldsymbol{h}_i^L + \boldsymbol{\delta}_i^L = \boldsymbol{W}_0^L \boldsymbol{k}_i^L + \boldsymbol{\delta}_i^L, \tag{7}$$

where $\boldsymbol{\delta}_i^L$ is a residual vector optimized by:

$$\mathbf{m}_i^L = \mathbf{h}_i^L + \arg\min_{\boldsymbol{\delta}_i^L} \frac{1}{P} \sum_{j=1}^{P} - \log \mathbb{P}_{\theta(\mathbf{h}_i^L += \boldsymbol{\delta}_i^L)}[o^* | x_j \oplus p] \tag{8}$$

The objective of the above equation is to optimize the learnable $\boldsymbol{\delta}_i^L$ to maximize the probability of the model predicting $o^*$. $x_j \oplus p$ represents the concatenation of the $j$th randomly generated $P$ prefixes by the model with prompt $p$ to enhance generalization. $\theta(\mathbf{h}_i^L += \boldsymbol{\delta}_i^L)$ denotes adding $\boldsymbol{\delta}_i^L$ to $\mathbf{h}_i^L$.

Fang et al. [6] extends locate-then-edit methods to the sequential batch editing scenario. They cache the keys $\boldsymbol{K}_p$ of previously edited knowledge and incorporate $\boldsymbol{K}_p$ into the least squares optimization, ultimately deriving the following closed-form solution for sequential batch editing:

$$\Delta_{\text{seq}}^l = \boldsymbol{R}^l \boldsymbol{K}_1^{l\,T} \left( \boldsymbol{K}_p^l \boldsymbol{K}_p^{l\,T} + \boldsymbol{K}_0^l \boldsymbol{K}_0^{l\,T} + \boldsymbol{K}_1^l \boldsymbol{K}_1^{l\,T} \right)^{-1} \tag{9}$$

## 4  Rethinking the Residual Distribution of Locate-then-Edit Model Editing

In this section, we analyze the residual distribution both empirically (Section 4.1) and theoretically (Section 4.2). Based on these insights, we further propose a novel strategy to enhance locate-then-edit model editing (Section 4.3). We focus on the classic locate-then-edit model editing method, MEMIT [5]. Experiments are conducted on three LLMs: Llama3-8B-Instruct [24], GPT-J (6B) [25], and GPT2-XL, using the CounterFact dataset [4]. Unless otherwise specified, we use the first 200 samples from the CounterFact dataset. The critical layers analyzed for each model are: Llama3-8B: $\{4, 5, 6, 7, 8\}$, GPT-J (6B): $\{3, 4, 5, 6, 7, 8\}$ and GPT2-XL: $\{13, 14, 15, 16, 17\}$.

### 4.1  Analyzing Residual Distribution in Locate-then-Edit Model Editing

#### 4.1.1  How Does the Distributed Residual Contribute to the Editing Object?

To measure the contribution of the distributed residuals to the editing object, we first define a **contribution score**:

$$s = \mathbb{P}_{\theta^*}(o^* | p) - \mathbb{P}_{\theta}(o^* | p) \tag{10}$$

where $\mathbb{P}_{\theta^*}(o^* | p)$ represents the probability of the post-edit model $\theta^*$ regarding the edited knowledge $t = (s, r, o) \rightarrow o^*$. The rationale behind this is that the probability of the pre-edit model $\theta$ assigning to $o^*$ on knowledge $t$ is often low, while the model editing aims for the post-edit model to assign the highest probability to $o^*$.

From Equ. (1) and Equ. (4), we can know that the essence of locate-then-edit is that the post-edit model can activate the new memory $\boldsymbol{m}_i^l = \boldsymbol{m}_{0,i}^l + \boldsymbol{\delta}_i^L/(L - l + 1)$ in the FFN block at layer $l$ using the key $\boldsymbol{k}_i^l$ corresponding to $p$, where $\boldsymbol{m}_{0,i}^l$ represents the original memory of the model. Therefore, we directly replace the output of the FFN block at layer $l$ with the new memory $\boldsymbol{m}_i^l$ to eliminate the potential impact of activation failure. For comparison, we also directly compute residuals for each layer, following the same process as $\boldsymbol{m}_i^L$. By using the distributed residual for *simulated editing*, we can accurately measure the contribution of new memories while avoiding direct edits to the model.

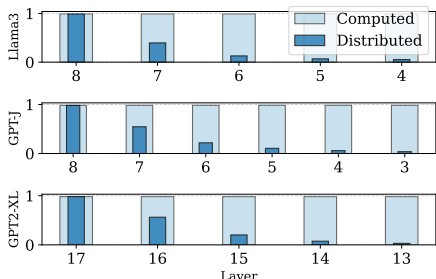

Figure 2: The average contribution score of different *simulated editing* layers.

We perform *simulated editing* in each critical layer. The average contribution scores are shown in Figure 2. For the distributed residuals, it can be observed that only the last critical layer achieves a contribution score close to $1.0$. The contribution scores of the other layers were all below $0.7$, showing a decreasing trend layer by layer. For the first critical layer, the contribution score is below $0.1$ in three LLMs. This indicates that **the farther the residuals are distributed, the lower their contribution to the editing object. Even distribute through just one layer can lead to a significant drop in the contribution score.** In contrast, for the computed residuals, their contribution scores in each layer consistently approach $1.0$.

#### 4.1.2 Is the Distributed Residual the Optimal Residual for Editing?

From Section 4.1.1, we observe that directly computing $m_i^l$ for each layer achieves high contribution scores. Therefore, we assume that the directly computed $m_i^l$ represents the optimal memory for editing. To verify whether residual distribution is optimal, we first compare the similarity between residual distribution and the directly computed $m_i^l$, and then evaluate their performance in model editing.

**Similarity Analyzing.** The variation in cosine similarity between the distributed and the directly computed $m_i^l$ is shown in Figure 3. It shows that the cosine similarity between the distributed and the directly computed $m_i^l$ exhibits a layer-by-layer decreasing trend, indicating that **the further residuals are distributed, the farther $m_i^l$ deviates from the optimal memory.** To further investigate how residual distribution affects model editing performance, we next perform single-layer model editing using both the distributed residuals and the directly computed residuals.

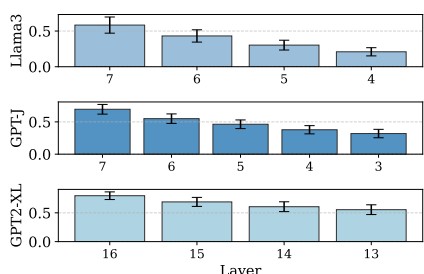

Figure 3: The variation in cosine similarity between the distributed and the directly computed memory across different layers.

**Post-edit LLM Performance.** We update single layers of the model using distributed residuals and computed residuals, respectively. The results for Llama3 under the batch editing setting are shown in Figure 4, while results for GPT-J and GPT2-XL are presented in Figure 9 of Appendix C. Specificity and Fluency remain comparable across different cases, with outcomes closely matching the original model. This indicates that small-batch edits effectively retain the model's original state. However, significant differences arise in Efficacy and Generalization between the two methods. For Efficacy, models edited with computed residuals outperform those with distributed residuals by over $3\times$ on average, while for Generalization, the improvement exceeds $2\times$. In terms of Consistency, computed residuals achieve an average improvement of more than 10%. **These findings indicate that distributed residuals introduce significant information loss during model editing, leading to a higher likelihood of editing failures.**

Figure 4: Performance variations when editing different single layers of the model using computed and distributed residuals separately. Fluency and Consistency are normalized.

### 4.2 Theoretical Analysis of Residual Distribution in Locate-then-Edit Methods

**Theorem 4.1.** *In the locate-then-edit model editing, when using residual distribution, the upper bound for the weight shift error between the exact weight shift $\Delta^{l^*}$ and the actual weight shift $\Delta^l$ is*

*given by*

$$\|\Delta^{l^*} - \Delta^l\|_2 \leq \left(\|\boldsymbol{R}^{l^*} - \boldsymbol{R}^L\|_2 + (L - l)\|\boldsymbol{R}^L\|_2\right)\|\boldsymbol{Q}\|_2, \tag{11}$$

*where $\boldsymbol{R}^{l^*}$ denotes the exact residual, and $\boldsymbol{Q} = \boldsymbol{K}_1^{l\,T}\left(\boldsymbol{K}_0^l\boldsymbol{K}_0^{l\,T} + \boldsymbol{K}_1^l\boldsymbol{K}_1^{l\,T}\right)^{-1}$.*

The proof of the above theorem is presented in Appendix A. We also discuss the rationale for using the upper bound instead of the lower bound in Remark 4.2. In Equ. (11), $\|\boldsymbol{R}^l\|_2$ and $\|\boldsymbol{Q}\|_2$ increase with the number of new memories (i.e., the size of the editing batch [5]). When the number of new memories is fixed, the upper bound increases with $\|\boldsymbol{R}^{l^*} - \boldsymbol{R}^L\|_2$ and $L - l$. $L - l$ increases as the residual distributes further, while it is unclear how $\|\boldsymbol{R}^{l^*} - \boldsymbol{R}^L\|_2$ changes. To explore this, we assume the computed residual is the exact residual and analyze how $\|\boldsymbol{R}^{l^*} - \boldsymbol{R}^L\|_2$ changes across layers. In Figure 5, we show how $\|\boldsymbol{R}^{l^*} - \boldsymbol{R}^L\|_2$ changes across layers. It shows that $\|\boldsymbol{R}^{l^*} - \boldsymbol{R}^L\|_2$ increases as the residual distribution extends farther. Therefore, we can conclude that **weight shift error increases with both the distance of residual distribution and the size of the editing batch**.

**Remark 4.2.** *The lower bound of the weight shift error can more effectively reflect the variation in error, but according to $\|\mathbf{R}^{l^*}\mathbf{Q} - \mathbf{R}^l\mathbf{Q}\|_2 \geq \sigma_{\min}(\mathbf{Q})\|\mathbf{R}^{l^*} - \mathbf{R}^l\|_2$ we see that the lower bound of the error shift is very small or even zero, where $\sigma_{\min}(\mathbf{Q})$ is the smallest singular value of $\mathbf{Q}$. This occurs because we cannot guarantee that $\mathbf{Q}$ is nondegenerate. Even if $\mathbf{Q}$ is nondegenerate, $\sigma_{\min}(\mathbf{Q})$ would still be very small, making the result insignificant. Additionally, although a growing upper bound does not necessarily imply an increase in error, our upper bound determines the worst-case scenario of the error, providing essential insight into selecting the layers to update.*

Considering the closed-form solution of locate-then-edit model editing in sequential batch editing, the following lemma can be derived.

**Lemma 4.3.** *In sequential batch editing, when using residual distribution, the upper bound of the weight shift error between the exact weight shift $\Delta^{l^*}$ and the actual weight shift $\Delta^l$ for locate-then-edit methods is given by*

$$\|\Delta^{l^*} - \Delta^l\|_2 \leq \left(\|\boldsymbol{R}^{l^*} - \boldsymbol{R}^L\|_2 + (L - l)\|\boldsymbol{R}^l\|_2\right)\|\boldsymbol{Q}'\|_2, \tag{12}$$

*where $\boldsymbol{R}^{l^*}$ denotes the exact residual, and $\boldsymbol{Q}' = \boldsymbol{K}_1^{l\,T}\left(\boldsymbol{K}_p^l\boldsymbol{K}_p^{l\,T} + \boldsymbol{K}_0^l\boldsymbol{K}_0^{l\,T} + \boldsymbol{K}_1^l\boldsymbol{K}_1^{l\,T}\right)^{-1}$.*

$\|\boldsymbol{K}_p^l\boldsymbol{K}_p^{l\,T}\|_2$ increases with the number of sequential edits, and thus Section 4.3 indicates that **the weight shift error also increases with the number of sequential edits.**

### 4.3 BLUE: Boundary Layer UpdatE for Improving Locate-then-Edit Model Editing

From the previous analysis, we know that the residual distribution of the locate-then-edit model editing is inherently inaccurate, and this inaccuracy increases as the residuals are distributed farther. So, *how can we enable locate-then-edit model editing to perform multi-layer updates while mitigating the negative impact of the residual distribution?* A straightforward method is to compute residuals separately for each layer. However, this reduces the efficiency of the locate-then-edit approach, and it remains unclear whether it is necessary to compute residuals and perform updates for all critical layers individually.

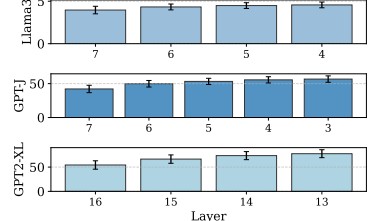

Figure 5: Variation of $\|\boldsymbol{R}^{l^*} - \boldsymbol{R}^L\|_2$ across layers.

Therefore, we conduct experiments where residuals are computed and updates performed for all critical layers. Computation is stopped when the loss falls below 0.05. We update the critical layers sequentially in the order of increasing layers and record the number of optimization steps required to compute residuals for each layer. The

Table 1: Average optimization steps.

| Model | **Layer**: Steps |
|---|---|
| GPT2-XL | **[13-17]**: [16.37, 8.43, 1.71, 0.32, 0.10] |
| GPT-J (6B) | **[3-8]**: [10.47, 1.68, 0.11, 0.0, 0.0, 0.0] |
| Llama3 (8B) | **[4-8]**: [25.0, 11.10, 0.63, 0.0, 0.0] |

results are shown in Table 1. It can be observed that after completing the first layer update, the number of residual computation steps in subsequent layers decreased significantly across all LLMs. In GPT2-XL, the decrease is 48.5%; in GPT-J, 84.0%; and in Llama3, 55.6%. Furthermore, after updating the first two layers, the optimization steps in the third layer for GPT2-XL, GPT-J and Llama3 drop below 2.0, indicating that **only two layers of weights needed to be updated to achieve the editing goal.**

The above observations indicate that when calculating residuals separately for all layers, updating just two layers is sufficient to achieve the editing object. The new question is: *which two layers should be updated to achieve optimal editing performance?* According to Theorem 4.1, the farther the distribution of residuals, the greater the upper bound of the weight shift error. The last critical layer is exactly the layer where the residual is computed. As a result, the first layer updated by the current method is the most affected by the residual distribution. To mitigate this effect, **we choose the first critical layer as the first layer for updates. For the second layer, we choose the last critical layer** for the following reasons: 1) The selection of the first critical key layer is based on the premise that the residual is computed at the last critical layer; and 2) BLUE is an optimization strategy, and we aim to preserve the mechanism used in existing locate-then-edit model editing methods, which compute residuals at the last key layer, to ensure broader applicability.

Therefore, we propose a **Boundary Layer UpdatE (BLUE)** strategy to boost the locate-then-edit methods. BLUE updates only the boundary layers of the critical layers by directly computing residuals of them, specifically the first critical layer and the last critical layer. This not only reduces the number of layers to be updated, but we also demonstrate in Section 5 that it performs better and better preserves LLMs' general capabilities. BLUE is suitable for locate-then-edit methods that perform multi-layer updates using even residual distribution: MEMIT [5], RECT [7], PRUNE [8] and AlphaEdit [6].

# 5 Experiments

In our experiments, we demonstrate that BLUE can enhance the performance of current locate-then-edit methods (Section 5.2), improve the retention of the original LLMs' capabilities (Section 5.3), and alleviate the hidden state shifts introduced by locate-then-edit approaches (Section 5.4). Additionally, in Appendix 5.5, we show that BLUE also boosts locate-then-edit methods in long-form model editing. In Appendix 5.6, we present ablation studies demonstrating that selecting the first and last critical layers for editing in BLUE is the optimal choice. Appendix I illustrates the efficiency improvements introduced by BLUE. In Appendix J, we empirically validate Theorem 4.1 and Lemma 4.3, demonstrating that the weight shift error increases with both the batch size and the number of sequential edits. Finally, we present the results of BLUE applied to the square root residual distribution method, PMET, in Appendix K.

## 5.1 Experimental Setup

**Datasets & LLMs.** Our experiments are conducted on two datasets: CounterFact [4] and zsRE [26]. We select three LLMs as the editing subjects: GPT2-XL [27], GPT-J (6B) [25], Llama3 (8B) [28]. We also conduct experiments with Llama2 (13B) to further validate the effectiveness of BLUE in Appendix G.

**Baselines.** BLUE is a facilitation strategy designed for locate-then-edit model editing that performs multi-layer updates, which has been proven in prior research to achieve the best editing performance [6]. Therefore, our baselines only consider locate-then-edit model editing methods. The locate-then-edit methods we consider are: MEMIT [5], PRUNE [8], RECT [29], and AlphaEdit [6]. We present the experimental details in the Appendix D.

## 5.2 Enhancing Editing Performance with BLUE

We first verify whether BLUE can enhance locate-then-edit model editing. Sequential batch editing better aligns with real-world batch knowledge updates, and we follow [6] by using sequential batch editing experiments to validate the capabilities of BLUE. We randomly sample 2,000 samples from the dataset and perform sequential batch editing with a batch size of 100. The results of the sequential batch editing are shown in Table 2. We use red to highlight the results enhanced by BLUE. The

Table 2: Comparison of BLUE enhanced locate-then-model editing methods with original locate-then-model editing methods on the sequential model editing task. We color all results that are actually enhanced by BLUE in red. We directly adopt the baseline results from [6] to avoid the energy consumption caused by redundant computation.

| Method | Model | Counterfact | | | | | ZsRE | | |
|---|---|---|---|---|---|---|---|---|---|
| | | Efficacy↑ | Generalization↑ | Specificity↑ | Fluency↑ | Consistency↑ | Efficacy↑ | Generalization↑ | Specificity↑ |
| Pre-edited | | $7.85_{\pm 0.26}$ | $10.58_{\pm 0.26}$ | $89.48_{\pm 0.18}$ | $635.23_{\pm 0.11}$ | $24.14_{\pm 0.08}$ | $36.99_{\pm 0.30}$ | $36.34_{\pm 0.30}$ | $31.89_{\pm 0.22}$ |
| MEMIT | Llama3 | $65.65_{\pm 0.47}$ | $64.65_{\pm 0.42}$ | $51.56_{\pm 0.38}$ | $437.43_{\pm 1.67}$ | $6.58_{\pm 0.11}$ | $34.62_{\pm 0.36}$ | $31.28_{\pm 0.34}$ | $18.49_{\pm 0.19}$ |
| PRUNE | | $68.25_{\pm 0.46}$ | $64.75_{\pm 0.41}$ | $49.82_{\pm 0.36}$ | $418.03_{\pm 1.52}$ | $5.90_{\pm 0.10}$ | $24.77_{\pm 0.27}$ | $23.87_{\pm 0.27}$ | $20.69_{\pm 0.23}$ |
| RECT | | $66.05_{\pm 0.47}$ | $63.62_{\pm 0.43}$ | $61.41_{\pm 0.37}$ | $526.62_{\pm 0.44}$ | $20.54_{\pm 0.09}$ | $86.05_{\pm 0.23}$ | $80.54_{\pm 0.27}$ | $31.67_{\pm 0.22}$ |
| AlphaEdit | | $98.90_{\pm 0.10}$ | $94.22_{\pm 0.19}$ | $67.88_{\pm 0.29}$ | $622.49_{\pm 0.16}$ | $32.40_{\pm 0.11}$ | $94.47_{\pm 0.13}$ | $91.13_{\pm 0.19}$ | $32.55_{\pm 0.22}$ |
| MEMIT_BLUE | | $99.57_{\pm 0.24}$ | $94.13_{\pm 0.77}$ | $83.77_{\pm 0.77}$ | $626.26_{\pm 0.51}$ | $32.29_{\pm 0.38}$ | $95.94_{\pm 0.38}$ | $90.98_{\pm 0.69}$ | $32.41_{\pm 0.81}$ |
| PRUNE_BLUE | | $96.73_{\pm 0.64}$ | $89.68_{\pm 0.85}$ | $57.79_{\pm 1.14}$ | $627.39_{\pm 0.37}$ | $33.39_{\pm 0.36}$ | $86.55_{\pm 0.86}$ | $82.22_{\pm 1.00}$ | $31.04_{\pm 0.80}$ |
| RECT_BLUE | | $98.77_{\pm 0.40}$ | $93.40_{\pm 0.74}$ | $79.34_{\pm 0.86}$ | $619.07_{\pm 0.62}$ | $30.62_{\pm 0.37}$ | $94.37_{\pm 0.49}$ | $89.49_{\pm 0.76}$ | $32.76_{\pm 0.81}$ |
| AlphaEdit_BLUE | | $99.93_{\pm 0.09}$ | $97.25_{\pm 0.48}$ | $75.24_{\pm 0.98}$ | $624.90_{\pm 0.49}$ | $33.79_{\pm 0.38}$ | $95.77_{\pm 0.39}$ | $91.73_{\pm 0.65}$ | $31.96_{\pm 0.80}$ |
| Pre-edited | | $16.22_{\pm 0.31}$ | $18.56_{\pm 0.45}$ | $83.11_{\pm 0.13}$ | $621.81_{\pm 0.67}$ | $29.74_{\pm 0.51}$ | $26.32_{\pm 0.37}$ | $25.79_{\pm 0.25}$ | $27.42_{\pm 0.53}$ |
| MEMIT | GPT-J | $98.55_{\pm 0.11}$ | $95.50_{\pm 0.16}$ | $63.64_{\pm 0.31}$ | $546.28_{\pm 0.88}$ | $34.89_{\pm 0.15}$ | $94.91_{\pm 0.16}$ | $90.22_{\pm 0.23}$ | $30.39_{\pm 0.27}$ |
| PRUNE | | $86.15_{\pm 0.34}$ | $86.85_{\pm 0.29}$ | $53.87_{\pm 0.35}$ | $427.14_{\pm 0.53}$ | $14.78_{\pm 0.11}$ | $0.15_{\pm 0.02}$ | $0.15_{\pm 0.02}$ | $0.00_{\pm 0.00}$ |
| RECT | | $98.80_{\pm 0.10}$ | $86.58_{\pm 0.28}$ | $72.22_{\pm 0.28}$ | $617.31_{\pm 0.19}$ | $41.39_{\pm 0.12}$ | $96.38_{\pm 0.14}$ | $91.21_{\pm 0.21}$ | $27.79_{\pm 0.26}$ |
| AlphaEdit | | $99.75_{\pm 0.08}$ | $96.38_{\pm 0.23}$ | $75.48_{\pm 0.21}$ | $618.50_{\pm 0.17}$ | $42.08_{\pm 0.15}$ | $99.79_{\pm 0.14}$ | $96.00_{\pm 0.22}$ | $28.29_{\pm 0.25}$ |
| MEMIT_BLUE | | $99.70_{\pm 0.30}$ | $96.90_{\pm 0.50}$ | $74.61_{\pm 0.95}$ | $620.89_{\pm 0.73}$ | $40.82_{\pm 0.44}$ | $99.58_{\pm 0.18}$ | $94.77_{\pm 0.67}$ | $28.36_{\pm 0.94}$ |
| PRUNE_BLUE | | $97.77_{\pm 0.53}$ | $97.28_{\pm 0.48}$ | $57.12_{\pm 1.00}$ | $608.73_{\pm 0.89}$ | $36.62_{\pm 0.42}$ | $60.51_{\pm 1.35}$ | $58.57_{\pm 1.35}$ | $22.77_{\pm 0.87}$ |
| RECT_BLUE | | $98.70_{\pm 0.41}$ | $91.18_{\pm 0.84}$ | $74.78_{\pm 0.94}$ | $620.52_{\pm 0.65}$ | $39.79_{\pm 0.43}$ | $97.93_{\pm 0.38}$ | $93.86_{\pm 0.69}$ | $26.32_{\pm 0.91}$ |
| AlphaEdit_BLUE | | $99.77_{\pm 0.17}$ | $97.13_{\pm 0.48}$ | $75.23_{\pm 0.95}$ | $621.07_{\pm 0.62}$ | $41.34_{\pm 0.44}$ | $99.63_{\pm 0.16}$ | $95.96_{\pm 0.59}$ | $28.67_{\pm 0.94}$ |
| Pre-edited | | $22.23_{\pm 0.73}$ | $24.34_{\pm 0.62}$ | $78.53_{\pm 0.33}$ | $626.64_{\pm 0.31}$ | $31.88_{\pm 0.20}$ | $22.19_{\pm 0.24}$ | $31.30_{\pm 0.27}$ | $24.15_{\pm 0.32}$ |
| MEMIT | GPT2-XL | $94.70_{\pm 0.22}$ | $85.82_{\pm 0.28}$ | $60.50_{\pm 0.32}$ | $477.26_{\pm 0.54}$ | $22.72_{\pm 0.15}$ | $79.17_{\pm 0.32}$ | $71.44_{\pm 0.36}$ | $26.42_{\pm 0.25}$ |
| PRUNE | | $82.05_{\pm 0.38}$ | $78.55_{\pm 0.34}$ | $53.02_{\pm 0.35}$ | $530.47_{\pm 0.39}$ | $15.93_{\pm 0.11}$ | $21.62_{\pm 0.30}$ | $19.27_{\pm 0.28}$ | $13.19_{\pm 0.18}$ |
| RECT | | $92.15_{\pm 0.26}$ | $81.15_{\pm 0.33}$ | $65.13_{\pm 0.31}$ | $480.83_{\pm 0.62}$ | $21.05_{\pm 0.16}$ | $81.02_{\pm 0.31}$ | $73.08_{\pm 0.35}$ | $24.85_{\pm 0.25}$ |
| AlphaEdit | | $99.50_{\pm 0.24}$ | $93.95_{\pm 0.34}$ | $66.39_{\pm 0.31}$ | $597.88_{\pm 0.18}$ | $39.38_{\pm 0.15}$ | $94.81_{\pm 0.30}$ | $86.11_{\pm 0.29}$ | $25.88_{\pm 0.21}$ |
| MEMIT_BLUE | | $98.27_{\pm 0.47}$ | $88.67_{\pm 0.93}$ | $67.13_{\pm 1.01}$ | $587.19_{\pm 1.52}$ | $35.64_{\pm 0.46}$ | $93.62_{\pm 0.70}$ | $85.34_{\pm 1.04}$ | $26.55_{\pm 0.93}$ |
| PRUNE_BLUE | | $88.19_{\pm 1.12}$ | $80.48_{\pm 1.10}$ | $52.23_{\pm 1.05}$ | $594.08_{\pm 1.18}$ | $20.28_{\pm 0.44}$ | $47.94_{\pm 1.33}$ | $45.03_{\pm 1.32}$ | $16.72_{\pm 0.75}$ |
| RECT_BLUE | | $95.67_{\pm 0.73}$ | $80.97_{\pm 1.15}$ | $66.88_{\pm 1.00}$ | $567.09_{\pm 2.09}$ | $30.30_{\pm 0.53}$ | $83.48_{\pm 1.06}$ | $75.24_{\pm 1.24}$ | $25.25_{\pm 0.89}$ |
| AlphaEdit_BLUE | | $99.40_{\pm 0.28}$ | $96.00_{\pm 0.60}$ | $76.63_{\pm 0.93}$ | $621.92_{\pm 0.56}$ | $40.98_{\pm 0.43}$ | $96.88_{\pm 0.50}$ | $89.58_{\pm 0.91}$ | $25.93_{\pm 0.92}$ |

results indicate that **the BLUE strategy effectively enhances the performance of a range of locate-then-edit methods in sequential batch editing tasks**. 89.58% of the results (86 out of 96) were enhanced. After using BLUE, the editing performance of different editing methods is enhanced across various LLMs, as shown in Table 4 of Appendix B. It can be observed that the BLUE strategy significantly improves the performance of PRUNE and noticeably enhances the editing performance of locate-then-edit methods on Llama3 and GPT2-XL. For other cases, such as AlphaEdit on GPT-J, the improvements are minimal due to its already strong baseline performance.

## 5.3 Boosting General Capability Retention via BLUE

Model editing should not affect other aspects of LLMs. In addition to using specificity and fluency for evaluation, this goal can also be achieved by assessing changes in the general capabilities of the models after editing. Following the work of [6], we evaluate the

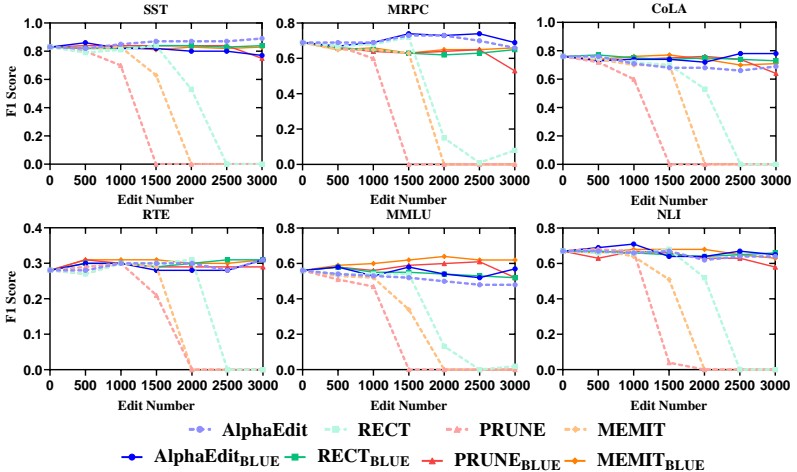

Figure 6: F1 scores of the post-edited Llama3 (8B) on six tasks.

general capabilities of LLMs before and after editing using six natural language tasks from the General Language Understanding Evaluation (GLUE) benchmark [30]. Specifically, we achieve this through the following six evaluation tasks: SST (The Stanford Sentiment Treebank) [31], MRPC (Microsoft Research Paraphrase Corpus) [32], MMLU (Massive Multi-task Language Understanding) [33], RTE (Recognizing Textual Entailment) [34], CoLA (Corpus of Linguistic Acceptability) [35], and NLI (Natural Language Inference) [36].

We conduct a total of 3,000 sequential edits on Llama3 (8B), with a batch size of 100 for each edit. Every 500 steps, we evaluate the performance of the post-edited LLMs on these six tasks. The results are shown in Figure 6. After 3,000 edits, the general capabilities of models edited by RECT, PRUNE, and MEMIT are almost entirely lost. In contrast, models edited by the BLUE-enhanced versions of these methods maintain their general capabilities well. Notably, AlphaEdit inherently demonstrates strong general capability retention, and AlphaEdit$_{BLUE}$ does not compromise this ability. These results indicate that **BLUE enhances the general capability retention of locate-then-edit methods**.

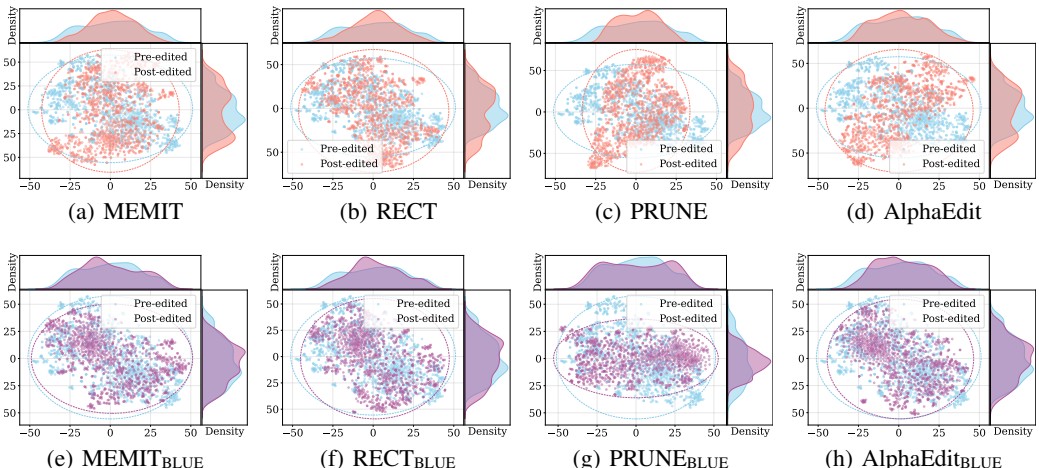

Figure 7: The distribution of hidden states in pre-edited and post-edited Llama3 (8B).

## 5.4 Mitigating Hidden States Shifts with BLUE

The existing locate-then-edit methods often result in shifts in the hidden states of the model after editing [6]. In this part, we verify whether BLUE can alleviate this phenomenon. Specifically, we extract the hidden states of 1,000 randomly selected factual prompts from LLMs before and after editing. These hidden states are then reduced to two dimensions using t-SNE. The post-editing LLMs mentioned here are the models described in Section 5.2. We then visualize the hidden states, and the results are shown in Figure 7. It can be observed that the shifts in hidden states corresponding to the locate-then-edit method enhanced by BLUE are weaker than those of the original method. This demonstrates that **BLUE can mitigate hidden states shifts caused by locate-then-edit methods**. The results on GPT-J (6B) and GPT2-XL can be found in Appendix H.

## 5.5 BLUE Boosts Long-Form Performance of Locate-Then-Edit Approaches

We conduct long-form model editing experiments on editing Llama3 (8B) using UnKEBench [10], a representative work in long-form evaluation benchmarks. We adopt MEMIT and AlphaEdit, both enhanced using the editing paradigm proposed in AnyEdit [9], as baselines, and follow the same experimental setup as used in AnyEdit. As illustrated in Table. 3, the results demonstrate that BLUE **can also improve the performance of locate-then-edit methods in long-form evaluation scenarios**. We also conduct a case study on a long-form editing task to demonstrate in detail the effectiveness of BLUE in Appendix E.

Table 3: Comparison of Editing Methods on UnKEBench. The left side of '/' represents the LLM's edited output for original questions, while the right side represents the edited output for paraphrase questions.

| Metric | BLEU | ROUGE-1 | ROUGE-2 | ROUGE-L | BERT Score | ROUGE-L (SubQ) |
|---|---|---|---|---|---|---|
| MEMIT | 77.65 / 66.17 | 90.78 / 81.60 | 87.12 / 73.94 | 90.43 / 80.81 | 95.69 / 92.32 | 47.36 |
| AlphaEdit | 62.82 / 51.78 | 80.02 / 69.08 | 71.62 / 56.80 | 79.10 / 67.68 | 91.67 / 87.41 | 42.72 |
| MEMIT$_{BLUE}$ | **82.34 / 74.76** | **90.79 / 85.36** | 86.89 / **78.94** | 90.33 / **84.68** | **96.50 / 94.72** | **52.15** |
| AlphaEdit$_{BLUE}$ | **68.64 / 61.49** | **84.90 / 78.54** | **77.98 / 68.40** | **84.08 / 77.37** | **93.39 / 90.97** | **49.49** |

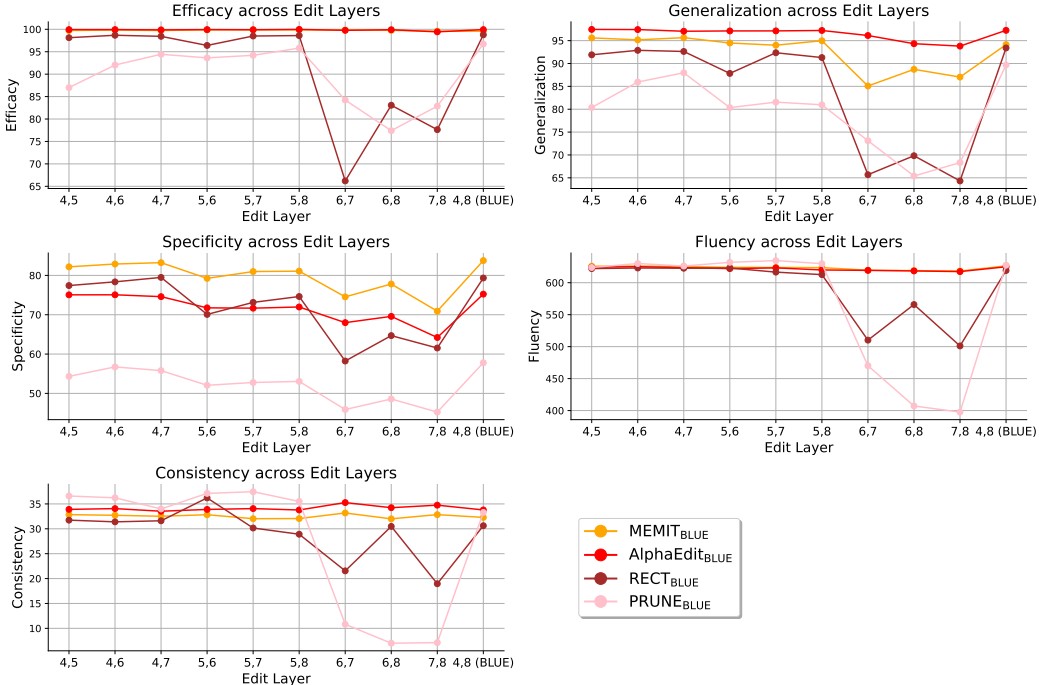

Figure 8: Layer Ablation of BLUE.

## 5.6 Ablation Study

To verify that the first and last critical layers selected by BLUE are optimal for editing, we perform 30 sequential edits with a batch size of 100 on Llama3 (8B). As shown in Figure 8, the optimal editing layers for all four BLUE-enhanced methods are consistently layers 4 and 8—the ones selected by BLUE. **This provides empirical evidence that the layers chosen by BLUE are indeed optimal for model editing.**

## 6 Conclusion

This paper rethinks the role of residual distribution in locate-then-edit model editing. Through empirical and theoretical analyses, we show that residual distribution is not an optimal choice, as it leads to increasing errors in weight updates with larger batch sizes, more sequential edits, and greater distribution distances. Based on these findings, we propose the BLUE strategy, which improves locate-then-edit methods by updating only the first and last critical layers of the model. Sequential batch editing experiments on three LLMs and two datasets demonstrate that BLUE effectively enhances editing performance. Further experiments and analyses indicate that BLUE also improves the retention of LLMs' original general capabilities and mitigates shifts in hidden states after editing. Moreover, BLUE yields gains in both time and memory efficiency and strengthens locate-then-edit methods in long-form model editing tasks.

## Acknowledgements

This work was supported by the National Key Research and Development Program of China under Grant 2024YFB4506200, the Science and Technology Innovation Program of Hunan Province under Grant 2024RC1048, the National Key Laboratory Foundation Project under Grant 2024-KJWPDL-14, and the National Natural Science Foundation of China under Grant 62402506.

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

# A Proof of Theorem 4.1

*Proof.* Let $\boldsymbol{K}_1^{l\,T} \left( \boldsymbol{K}_0^l \boldsymbol{K}_0^{l\,T} + \boldsymbol{K}_1^l \boldsymbol{K}_1^{l\,T} \right)^{-1} := \boldsymbol{Q}$, then the weight shifts error is:

$$\|\Delta^{l^*} - \Delta^l\|_2 = \|\boldsymbol{R}^{l^*}\boldsymbol{Q} - \boldsymbol{R}^l\boldsymbol{Q}\|_2 \tag{13}$$

$$\leq \|\boldsymbol{R}^{l^*} - \boldsymbol{R}^l\|_2 \|\boldsymbol{Q}\|_2 \tag{14}$$

$$\leq \|\boldsymbol{R}^{l^*} - \frac{\boldsymbol{R}^L}{L - l + 1}\|_2 \|\boldsymbol{Q}\|_2 \tag{15}$$

According to Section 4.1.2, the directly computed $\boldsymbol{m}_i^l$ represents the optimal memory for editing, and thus we have $\boldsymbol{R}^L = \boldsymbol{R}^{L^*}$. Then, Equ. (15) can be written as:

$$\|\boldsymbol{R}^{l^*} - \frac{\boldsymbol{R}^{L^*}}{L - l + 1}\|_2 \|\boldsymbol{Q}\|_2 \tag{16}$$

$$= \|\boldsymbol{R}^{l^*} - \boldsymbol{R}^{L^*} + \boldsymbol{R}^{L^*} - \frac{\boldsymbol{R}^{L^*}}{L - l + 1}\|_2 \|\boldsymbol{Q}\|_2 \tag{17}$$

$$\leq \left( \|\boldsymbol{R}^{l^*} - \boldsymbol{R}^{L^*}\|_2 + \|\boldsymbol{R}^{L^*} - \frac{\boldsymbol{R}^{L^*}}{L - l + 1}\|_2 \right) \|\boldsymbol{Q}\|_2 \tag{18}$$

$$\leq \left( \|\boldsymbol{R}^{l^*} - \boldsymbol{R}^{L^*}\|_2 + \frac{L - l}{L - l + 1}\|\boldsymbol{R}^{L^*}\|_2 \right) \|\boldsymbol{Q}\|_2 \tag{19}$$

$$\leq \left( \|\boldsymbol{R}^{l^*} - \boldsymbol{R}^{L}\|_2 + (L - l)\|\boldsymbol{R}^{L}\|_2 \right) \|\boldsymbol{Q}\|_2 \tag{20}$$

$\square$

# B BLUE's Performance Improvement in Locate-then-Edit Model Editing Across LLMs

We summarize the enhancement degree of each method by BLUE across different models, as shown in the table below.

Table 4: The average performance improvement of BLUE on different locate-then-edit model editing across various LLMs. The abnormal results of PRUNE editing GPT-J on the ZsRE dataset are excluded in our statistics.

| Model | MEMIT | PRUNE | RECT | AlphaEdit |
|---|---|---|---|---|
| Llama3 | 129.61% | 144.52% | 27.05% | 2.50 % |
| GPT-J | 6.73% | 44.36% | 0.58% | 0.03% |
| GPT2-XL | 17.02% | 41.24% | 9.47% | 4.00% |

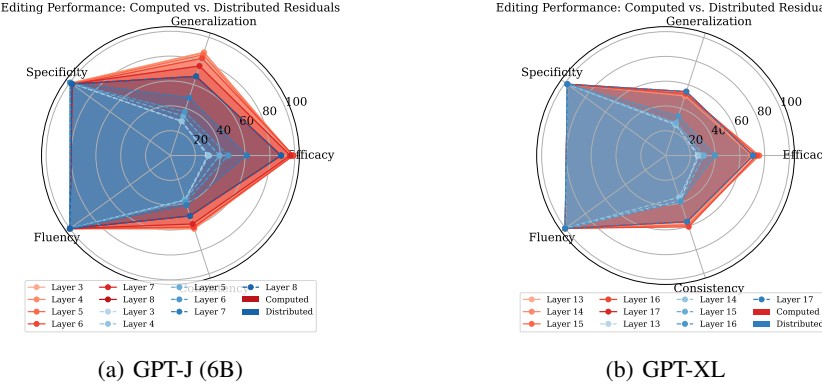

(a) GPT-J (6B)          (b) GPT-XL

Figure 9: Performance variations when editing different single layers of the model using computed and distributed residuals separately. For better visualization, Fluency and Consistency were normalized.

## C    Supplementary Results of Post-edit LLM Performance

We show the supplementary results of post-edit LLM performance in Figure 9. The results also indicate that distributed residuals introduce significant information loss during model editing, leading to a higher likelihood of editing failures.

## D    Experiment Details

All our experiments are conducted on A800 GPUs. The baseline methods used for comparison in the experiments are kept in their original settings, with PRUNE following the reproduction settings of [6]. For baseline methods enhanced by BLUE, all configurations remain consistent with the original baselines, except for AlphaEdit$_{\text{BLUE}}$. For AlphaEdit$_{\text{BLUE}}$, we set the $\alpha$ values for Llama3 (8B), GPT-J (6B), and GPT2-XL to 1, 95, and 80, respectively, to ensure the invertibility of matrices during the editing process, thereby achieving better editing performance. For a clearer understanding of the baselines, please refer to [6].

## E    Case Study on Long-form Model Editing Task

To demonstrate in detail the improvements of BLUE in long-form model editing, we present a case study of editing Llama-3 (8B) in Table 5. It can be seen that the method enhanced by BLUE generates content that is noticeably closer to the target output compared to the original method. This provides additional evidence for the effectiveness of BLUE in long-form model editing tasks.

## F    Batch Model Editing

In addition to sequential batch editing, large-scale batch editing is also an important aspect of evaluating the performance of model editing methods. Therefore, we conducted 10,000 batch edits for both the baseline and the BLUE-enhanced methods, with the results shown in Table 6. The results in the table indicate that while the improvement in large-scale batch editing after applying the BLUE enhancement to the baseline is not as significant as in sequential batch editing, the baselines enhanced by BLUE still demonstrate overall stronger performance. Specifically, 70.83% of the metrics (68 out of 96) are improved. Note that although the baselines enhanced by BLUE performed better in terms of efficacy and generalization, they show worse results in specificity. This suggests that while the BLUE-enhanced model editing methods strengthen the knowledge being edited, it also affects other unrelated knowledge. Achieving optimal performance across all three metrics simultaneously remains a major challenge in model editing [14]. This is particularly true for locate-then-edit methods, as BLUE serves as an enhancement to existing editing methods without altering their original modeling. Therefore, addressing this issue may require future work on improving the original modeling of

Table 5: Examples of different editing methods in long-form editing scenarios.

| Type | Text |
|---|---|
| Input | What is George Rankin's occupation? |
| Paraphrase | What does George Rankin do for a living? |
| Target Output | George Rankin has been actively involved in politics for over a decade. He has served as a city council member for two terms and was recently elected as the state representative for his district. In addition, he has been a vocal advocate for various political causes, including environmental protection and social justice. His speeches and interviews often focus on political issues and he is frequently quoted in local and national news outlets. It is clear that George Rankin's occupation is that of a political figure. |
| AlphaEdit | George Rankin has been involved in politics for over 20 years. He has served as a member of the state legislature for 10 years and has been a mayor for 5 years. He has also been a member of various political committees and has run for several political offices, including governor and U.S. Senate. |
| MEMIT | George Rankin has been involved in politics for over 20 years. He has served as a member of the state legislature for 10 years and has been a mayor for 5 years. He has also run for governor and senator, but has not held those offices. |
| AlphaEdit$_{BLUE}$ | George Rankin has been actively involved in politics for over a decade. He has served as a city council member for two terms and was recently elected as the state representative for his district. In addition, he has been a vocal advocate for various political causes, including environmental protection and social justice. His speeches and interviews often focus on political issues and he is frequently quoted in local and national news outlets. It is clear that George Rankin has been involved in politics for a long time and continues to be an active and influential figure in the political scene. |
| MEMIT$_{BLUE}$ | George Rankin has been actively involved in politics for over a decade. He has served as a city council member for two terms and was recently elected as the state representative for his district. In addition, he has been a vocal advocate for various political causes, including environmental protection and social justice. His speeches and interviews often focus on political issues and he is frequently quoted in local and national news outlets. It is clear that George Rankin's involvement in politics is a significant aspect of his public persona and identity. |

editing methods. Nevertheless, the overall superior performance of the BLUE-enhanced methods also demonstrates that BLUE can improve the editing performance of locate-then-edit approaches in batch editing.

## G   Results on LLaMA-2-13B

To demonstrate the effectiveness of BLUE on larger models, we selecte AlphaEdit and MEMIT to perform sequential editing experiments on layers 30–34 of LLaMA-2-13B, following the same setup as in the main experiments. The results are shown in Table 7. The results demonstrate that the BLUE-enhanced methods achieve better overall editing performance than the baseline methods on both the CounterFact and ZsRE datasets. We bolded all the enhanced results. On the CounterFact dataset, all metrics except for Specificity show improvements over the baseline. On the ZsRE dataset, all metrics of AlphaEdit show improvements, while MEMIT exhibits improvements in all metrics except for Specificity. This indicates that BLUE is also effective on LLaMA-2-13B, further validating the effectiveness of our approach.

## H   Hidden States Shifts in GPT-J (6B) and GPT2-XL

We present the hidden state shifts before and after model editing for GPT-J (6B) and GPT2-XL in Figs. 10 and 11, respectively. Similar results to those on Llama3 (8B) are observed for GPT-J (6B) and GPT-2 XL. The BLUE-enhanced baselines have a smaller overall impact on the model's hidden

Table 6: Comparison of BLUE enhanced locate-then-model editing methods with original locate-then-model editing methods on the batch model editing task. *Eff.*, *Gen.*, *Spe.*, *Flu.* and *Consis.* denote Efficacy, Generalization, Specificity, Fluency and Consistency, respectively. We color all results that are actually enhanced by BLUE in red.

| Method | Model | Counterfact | | | | | ZsRE | | |
|---|---|---|---|---|---|---|---|---|---|
| | | Eff.↑ | Gen.↑ | Spe.↑ | Flu.↑ | Consis.↑ | Eff.↑ | Gen.↑ | Spe.↑ |
| Pre-edited | | $7.02_{\pm0.50}$ | $9.44_{\pm0.49}$ | $89.73_{\pm0.36}$ | $630.00_{\pm0.22}$ | $24.21_{\pm0.17}$ | $35.67_{\pm0.58}$ | $34.81_{\pm0.58}$ | $31.83_{\pm0.44}$ |
| MEMIT | | $93.53_{\pm0.48}$ | $74.12_{\pm0.75}$ | $84.18_{\pm0.40}$ | $626.24_{\pm0.26}$ | $29.71_{\pm0.20}$ | $86.57_{\pm0.48}$ | $82.58_{\pm0.54}$ | $32.47_{\pm0.44}$ |
| PRUNE | | $93.64_{\pm0.48}$ | $84.44_{\pm0.57}$ | $60.00_{\pm0.57}$ | $625.11_{\pm0.26}$ | $36.83_{\pm0.22}$ | $13.29_{\pm0.34}$ | $13.75_{\pm0.52}$ | $15.34_{\pm0.54}$ |
| RECT | Llama3 | $58.07_{\pm0.97}$ | $39.88_{\pm0.86}$ | $88.15_{\pm0.37}$ | $628.71_{\pm0.23}$ | $26.11_{\pm0.18}$ | $70.35_{\pm0.65}$ | $65.04_{\pm0.67}$ | $32.45_{\pm0.44}$ |
| AlphaEdit | | $88.89_{\pm0.62}$ | $69.91_{\pm0.79}$ | $83.98_{\pm0.41}$ | $625.78_{\pm0.25}$ | $28.66_{\pm0.19}$ | $84.07_{\pm0.52}$ | $80.15_{\pm0.57}$ | $32.50_{\pm0.44}$ |
| MEMIT$_{BLUE}$ | | $99.28_{\pm0.17}$ | $93.83_{\pm0.40}$ | $79.34_{\pm0.46}$ | $626.09_{\pm0.26}$ | $33.04_{\pm0.21}$ | $95.38_{\pm0.23}$ | $92.62_{\pm0.33}$ | $31.68_{\pm0.44}$ |
| PRUNE$_{BLUE}$ | | $99.37_{\pm0.16}$ | $94.30_{\pm0.35}$ | $60.17_{\pm0.57}$ | $623.54_{\pm0.24}$ | $36.64_{\pm0.21}$ | $86.83_{\pm0.44}$ | $83.55_{\pm0.50}$ | $28.23_{\pm0.42}$ |
| RECT$_{BLUE}$ | | $94.02_{\pm0.46}$ | $79.25_{\pm0.69}$ | $85.45_{\pm0.39}$ | $627.57_{\pm0.24}$ | $30.34_{\pm0.19}$ | $85.27_{\pm0.48}$ | $77.73_{\pm0.58}$ | $31.83_{\pm0.44}$ |
| AlphaEdit$_{BLUE}$ | | $98.62_{\pm0.23}$ | $90.76_{\pm0.48}$ | $79.61_{\pm0.45}$ | $625.04_{\pm0.27}$ | $32.46_{\pm0.21}$ | $94.14_{\pm0.27}$ | $90.64_{\pm0.38}$ | $31.52_{\pm0.44}$ |
| Pre-edited | | $15.20_{\pm0.70}$ | $17.70_{\pm0.60}$ | $83.50_{\pm0.50}$ | $622.40_{\pm0.30}$ | $29.40_{\pm0.20}$ | $26.40_{\pm0.60}$ | $25.80_{\pm0.50}$ | $27.00_{\pm0.50}$ |
| MEMIT | | $98.72_{\pm0.22}$ | $87.14_{\pm0.56}$ | $74.05_{\pm0.52}$ | $620.68_{\pm0.30}$ | $39.72_{\pm0.24}$ | $95.90_{\pm0.30}$ | $89.06_{\pm0.49}$ | $26.30_{\pm0.50}$ |
| PRUNE | | $91.54_{\pm0.55}$ | $90.00_{\pm0.49}$ | $57.49_{\pm0.60}$ | $562.52_{\pm0.58}$ | $37.34_{\pm0.20}$ | $29.98_{\pm0.67}$ | $26.91_{\pm0.65}$ | $16.75_{\pm0.40}$ |
| RECT | GPT-J | $88.13_{\pm0.63}$ | $63.40_{\pm0.83}$ | $79.31_{\pm0.50}$ | $622.54_{\pm0.27}$ | $35.62_{\pm0.22}$ | $70.46_{\pm0.70}$ | $61.90_{\pm0.73}$ | $26.64_{\pm0.50}$ |
| AlphaEdit | | $99.26_{\pm0.17}$ | $86.70_{\pm0.56}$ | $69.65_{\pm0.53}$ | $587.89_{\pm0.49}$ | $39.51_{\pm0.23}$ | $93.09_{\pm0.36}$ | $82.64_{\pm0.59}$ | $22.78_{\pm0.47}$ |
| MEMIT$_{BLUE}$ | | $99.58_{\pm0.13}$ | $97.40_{\pm0.25}$ | $64.92_{\pm0.55}$ | $615.97_{\pm0.36}$ | $40.83_{\pm0.25}$ | $98.18_{\pm0.18}$ | $93.61_{\pm0.38}$ | $25.91_{\pm0.49}$ |
| PRUNE$_{BLUE}$ | | $99.36_{\pm0.16}$ | $98.06_{\pm0.22}$ | $56.82_{\pm0.55}$ | $608.78_{\pm0.33}$ | $41.76_{\pm0.22}$ | $74.83_{\pm0.65}$ | $71.24_{\pm0.69}$ | $20.07_{\pm0.46}$ |
| RECT$_{BLUE}$ | | $97.86_{\pm0.28}$ | $88.41_{\pm0.54}$ | $74.91_{\pm0.52}$ | $621.45_{\pm0.30}$ | $38.79_{\pm0.23}$ | $90.20_{\pm0.46}$ | $81.00_{\pm0.61}$ | $27.31_{\pm0.51}$ |
| AlphaEdit$_{BLUE}$ | | $99.39_{\pm0.15}$ | $95.52_{\pm0.35}$ | $69.28_{\pm0.54}$ | $619.90_{\pm0.32}$ | $41.25_{\pm0.24}$ | $98.26_{\pm0.19}$ | $96.63_{\pm0.39}$ | $26.59_{\pm0.50}$ |
| Pre-edited | | $21.82_{\pm0.81}$ | $24.16_{\pm0.72}$ | $78.32_{\pm0.55}$ | $626.78_{\pm0.23}$ | $31.37_{\pm0.20}$ | $22.17_{\pm0.52}$ | $21.28_{\pm0.51}$ | $24.2_{\pm0.48}$ |
| MEMIT | | $79.64_{\pm0.79}$ | $65.86_{\pm0.83}$ | $70.01_{\pm0.56}$ | $625.67_{\pm0.27}$ | $36.17_{\pm0.22}$ | $62.46_{\pm0.75}$ | $57.59_{\pm0.77}$ | $25.86_{\pm0.50}$ |
| PRUNE | | $85.27_{\pm0.69}$ | $78.30_{\pm0.70}$ | $57.73_{\pm0.62}$ | $604.09_{\pm0.39}$ | $35.66_{\pm0.21}$ | $42.71_{\pm0.76}$ | $40.14_{\pm0.75}$ | $19.01_{\pm0.44}$ |
| RECT | GPT2-XL | $61.92_{\pm0.95}$ | $48.68_{\pm0.87}$ | $74.69_{\pm0.54}$ | $625.87_{\pm0.25}$ | $33.99_{\pm0.21}$ | $49.37_{\pm0.76}$ | $45.30_{\pm0.74}$ | $25.64_{\pm0.49}$ |
| AlphaEdit | | $93.24_{\pm0.49}$ | $76.28_{\pm0.71}$ | $64.54_{\pm0.57}$ | $604.70_{\pm0.38}$ | $38.62_{\pm0.23}$ | $61.26_{\pm0.74}$ | $54.82_{\pm0.76}$ | $20.83_{\pm0.45}$ |
| MEMIT$_{BLUE}$ | | $87.54_{\pm0.65}$ | $78.14_{\pm0.71}$ | $65.37_{\pm0.54}$ | $615.34_{\pm0.43}$ | $37.10_{\pm0.22}$ | $71.93_{\pm0.72}$ | $67.51_{\pm0.75}$ | $23.44_{\pm0.49}$ |
| PRUNE$_{BLUE}$ | | $95.70_{\pm0.40}$ | $90.18_{\pm0.48}$ | $53.81_{\pm0.57}$ | $596.30_{\pm0.56}$ | $37.02_{\pm0.24}$ | $51.06_{\pm0.77}$ | $47.82_{\pm0.76}$ | $14.74_{\pm0.39}$ |
| RECT$_{BLUE}$ | | $70.93_{\pm0.89}$ | $58.73_{\pm0.85}$ | $72.09_{\pm0.53}$ | $621.52_{\pm0.34}$ | $34.78_{\pm0.21}$ | $58.88_{\pm0.77}$ | $54.29_{\pm0.77}$ | $24.55_{\pm0.49}$ |
| AlphaEdit$_{BLUE}$ | | $94.10_{\pm0.46}$ | $81.20_{\pm0.65}$ | $64.53_{\pm0.56}$ | $620.79_{\pm0.31}$ | $38.81_{\pm0.21}$ | $76.68_{\pm0.65}$ | $70.16_{\pm0.73}$ | $23.00_{\pm0.47}$ |

Table 7: Comparison of different editing methods on CounterFact and ZsRE datasets when editing LLaMA-2-13B.

| Method | CounterFact Dataset | | | | | ZsRE Dataset | | |
|---|---|---|---|---|---|---|---|---|
| | Efficacy | Generalization | Specificity | Fluency | Consistency | Efficacy | Generalization | Specificity |
| AlphaEdit | $52.30_{\pm2.19}$ | $48.95_{\pm1.73}$ | $\mathbf{50.53}_{\pm2.21}$ | $408.92_{\pm2.36}$ | $0.22_{\pm0.01}$ | $52.31_{\pm1.44}$ | $37.68_{\pm1.49}$ | $9.11_{\pm0.60}$ |
| AlphaEdit$_{BLUE}$ | $\mathbf{78.75}_{\pm1.79}$ | $\mathbf{69.53}_{\pm1.81}$ | $44.27_{\pm1.76}$ | $\mathbf{513.49}_{\pm3.09}$ | $\mathbf{22.10}_{\pm0.57}$ | $\mathbf{53.01}_{\pm2.32}$ | $\mathbf{41.32}_{\pm1.52}$ | $\mathbf{9.18}_{\pm0.60}$ |
| MEMIT | $79.20_{\pm1.77}$ | $65.94_{\pm1.79}$ | $\mathbf{41.53}_{\pm1.75}$ | $378.10_{\pm4.01}$ | $14.44_{\pm0.59}$ | $50.11_{\pm1.46}$ | $34.70_{\pm1.47}$ | $\mathbf{9.71}_{\pm0.61}$ |
| MEMIT$_{BLUE}$ | $\mathbf{79.60}_{\pm1.78}$ | $\mathbf{66.00}_{\pm1.81}$ | $41.28_{\pm1.65}$ | $\mathbf{384.23}_{\pm3.50}$ | $\mathbf{14.65}_{\pm0.59}$ | $\mathbf{51.07}_{\pm1.45}$ | $\mathbf{37.13}_{\pm1.52}$ | $9.07_{\pm0.60}$ |

states compared to the original baselines. This indicates that BLUE can mitigate the hidden state shifts caused by locate-then-edit methods, suggesting that **the BLUE-enhanced baselines introduce fewer side effects to the original model.**

# I    Efficiency Analysis

To evaluate the time and peak memory required for a single edit, we perform 300 sequential edits with a batch size of 1 and record the editing time and peak memory for each edit. The average peak memory consumption and editing time per instance for each method are reported in Table 8 and Figure 12. In terms of peak memory, BLUE achieves higher memory efficiency than the original methods. This is

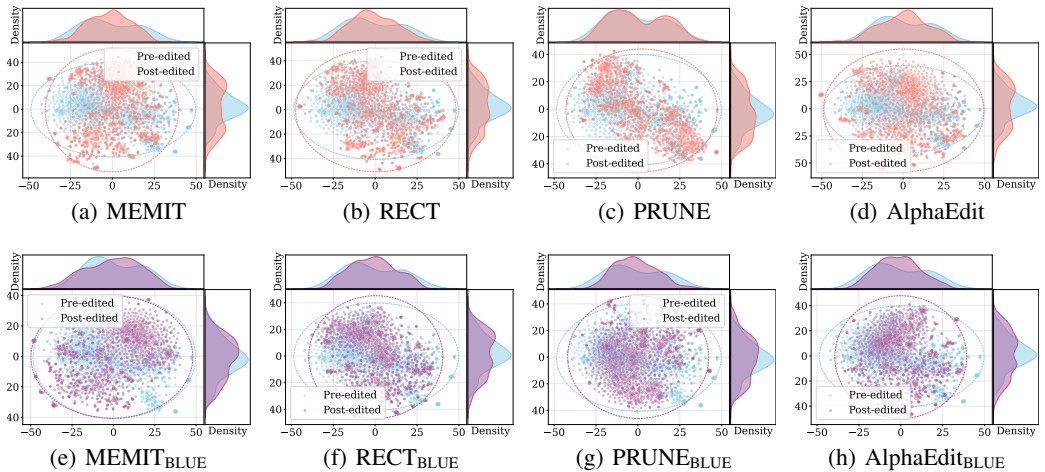

Figure 10: The distribution of hidden states in pre-edited and post-edited GPT-J (6B).

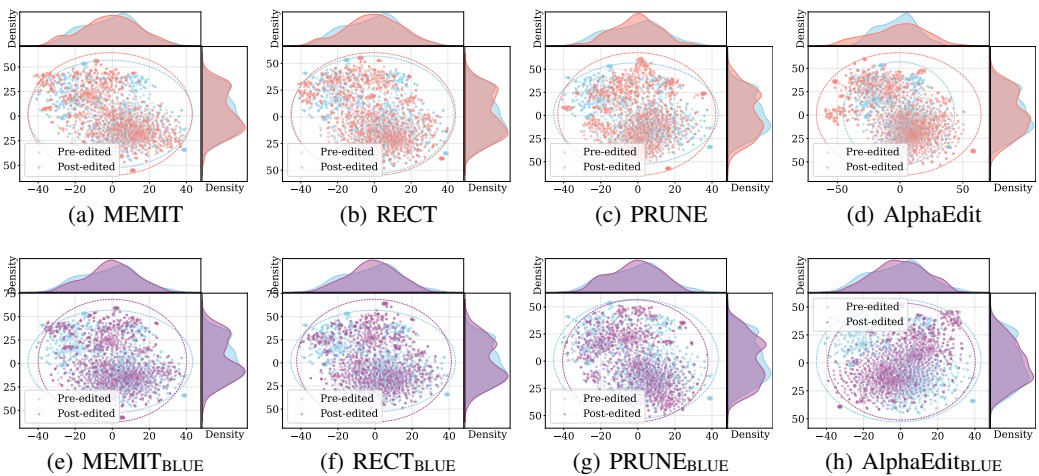

Figure 11: The distribution of hidden states in pre-edited and post-edited GPT2-XL.

intuitive: by discarding residual distributions and instead computing residuals only for the first and last key layers, BLUE reduces the memory overhead associated with residual distribution. Regarding editing time, with the exception of PRUNEBLUE on Llama-3 (8B)—which incurs a slightly higher editing time than its original counterpart—BLUE-enhanced methods consistently achieve lower average editing times across different models and datasets. These findings demonstrate that **BLUE not only improves the editing performance of locate-then-edit methods, but also enhances their editing efficiency.**

## J Experimental validation of the theorem and lemma

In this section, we experimentally validate the correctness of the theorem and lemma presented in Section 4.2. In this section, we choose llama3 (8B) as the target model for editing, and use MEMIT and AlphaEdit as the editing methods.

### J.1 Experimental verification of Theorem 4.1

According to Theorem 4.1, the weight shift error increases with the batch editing size. To verify this, we conduct batch editing experiments with 500, 1000, 5000, and 10000 edits on the CounterFact and ZsRE datasets. The results are shown in Figure 13. It can be clearly observed that on both datasets,

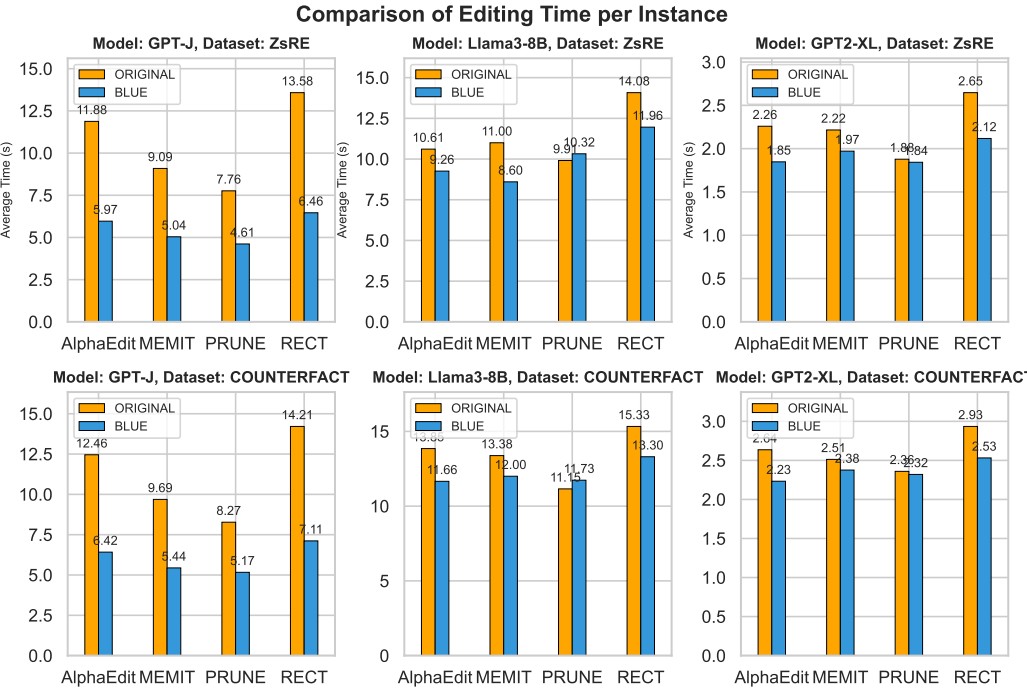

Figure 12: Average Editing Time per Instance for Different Methods

Table 8: Peak memory usage of both BLUE and locate-then-edit

| Method | CounterFact | | | ZsRE | | |
|---|---|---|---|---|---|---|
| | Llama-3 | GPT-J | GPT-2XL | Llama-3 | GPT-J | GPT-2XL |
| AlphaEdit | 36.09 | 30.43 | 7.14 | 36.09 | 30.43 | 7.43 |
| AlphaEdit$_{BLUE}$ | 35.10 | 28.92 | 7.24 | 35.43 | 28.92 | 7.49 |
| MEMIT | 37.07 | 31.68 | 7.34 | 37.07 | 31.68 | 7.34 |
| MEMIT$_{BLUE}$ | 36.42 | 30.67 | 7.30 | 36.42 | 30.67 | 7.38 |
| RECT | 37.07 | 31.68 | 7.34 | 37.07 | 31.68 | 7.34 |
| RECT$_{BLUE}$ | 36.42 | 30.67 | 7.22 | 36.42 | 30.67 | 7.37 |
| PRUNE | 38.17 | 33.18 | 7.53 | 38.17 | 33.18 | 7.53 |
| PRUNE$_{BLUE}$ | 36.85 | 30.67 | 7.30 | 36.85 | 30.67 | 7.46 |

the overall performance gap between BLUE-enhanced methods and non-enhanced methods increases with the batch size, especially in terms of efficacy and generalization. Although not all metrics show a strictly increasing performance gap with larger batch sizes—for example, on the Consistency metric of the CounterFact dataset, the performance gap for 5000 batch edits is greater than that for 10000 edits—the overall trend still empirically supports the correctness of Theorem 4.1.

## J.2 Experimental verification of Lemma 4.3

According to Lemma 4.3, the weight shift error increases with the number of sequential edits. To verify this, we conduct sequential editing experiments with a batch size of 1, using 100, 500, 1000, and 5000 edits. As shown in Figure 14, similar to the batch editing results, the overall trend on both datasets shows that the performance gap between the BLUE-enhanced methods and the original methods increases with the number of sequential edits. For instance, on the CounterFact dataset, the performance gap in efficacy and generalization between AlphaEdit and AlphaEdit$_{BLUE}$ increases significantly when the number of edits reaches 5000. Although specificity and consistency do not exhibit this trend, the patterns observed in efficacy and generalization still support the conclusion that weight shift error increases with the number of sequential edits, as efficacy and generalization directly reflect the impact of the editing methods.

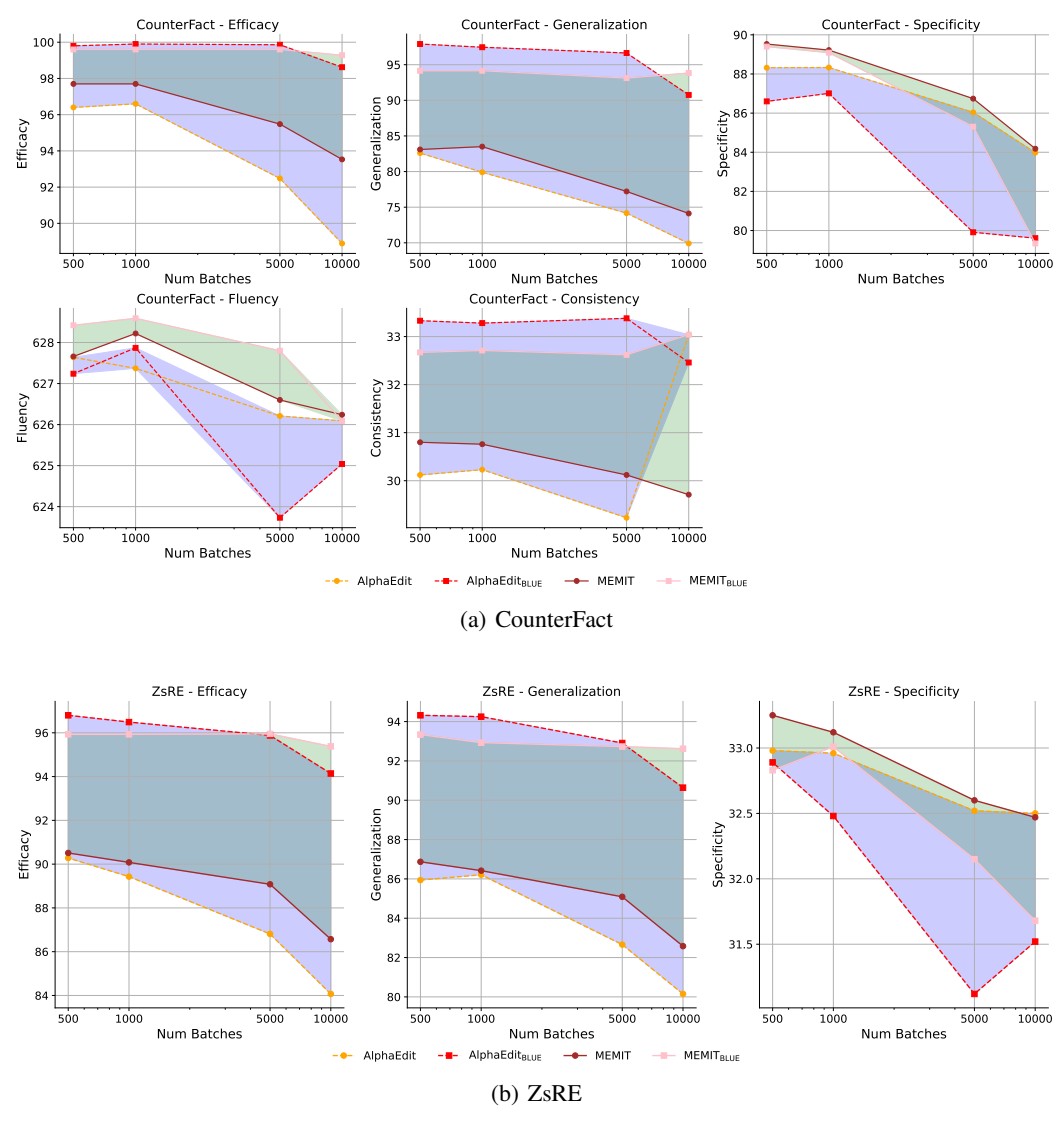

(a) CounterFact

(b) ZsRE

Figure 13: Performance Variation of Model Editing with Batch Edits on CounterFact and ZsRE Datasets

# K BLUE in the Locate-then-edit Method with Square Root Residual Distribution

Some locate-then-edit methods (e.g., PMET [21]) use a square root residual distribution instead of an even spread. They claim that the square root residual distribution can mitigate information loss during residual distribution. Since BLUE is designed for locate-then-edit methods with even residual distribution, we do not consider such methods as baselines. Nevertheless, we attempt to enhance PMET with BLUE. The results of sequential batch editing are shown in Table 9. PMET$_{BLUE}$ exhibits a significant performance improvement when editing Llama3 on sequential model editing task, while its performance gains in other scenarios are relatively limited. We speculate that this may be because PMET's use of square root distribution retains more editing information compared to even distribution, leading to the limited improvement of BLUE. Additionally, PMET incorporates a self-attention module during editing optimization but only edits the FFN weights when updating model parameters. This might result in BLUE's two-layer update being insufficient to fully integrate the editing information into the model weights. Nevertheless, BLUE demonstrates notable improvements

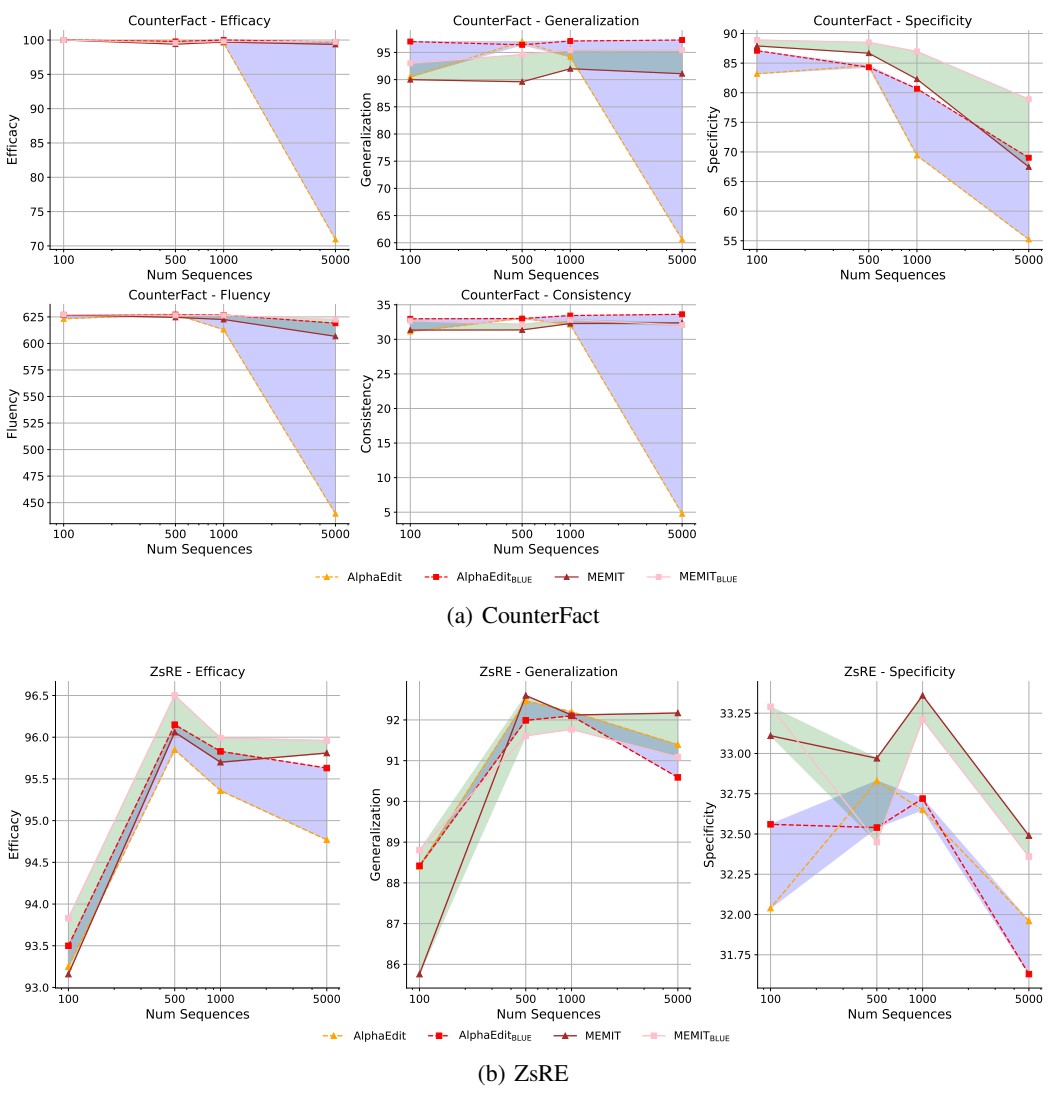

(a) CounterFact

(b) ZsRE

Figure 14: Performance Variation of Model Editing with Sequential Edits on CounterFact and ZsRE Datasets

in the locate-then-edit approaches with residual even distribution, indicating that it remains applicable to most locate-then-edit methods.

# L    Limitations

As a general strategy applicable to locate-then-edit methods, BLUE enhances various aspects of the locate-then-edit paradigm. Extensive evidence demonstrates that BLUE improves the editing performance of locate-then-edit methods, preserves the original model's capabilities post-editing, and alleviates the shift in hidden representations introduced by editing. Moreover, with updates applied to only two layers, it also improves editing efficiency. However, the improvement of BLUE in the locate-then-edit method for editing reasoning knowledge (e.g., MQuAKE [11]) remains to be verified, and we leave it as future work..

Table 9: Comparison of PMET$_{\text{BLUE}}$ with original PMET on the sequential batch and batch model editing task. We color all results that are actually enhanced by BLUE in red.

| Method | Model | Counterfact | | | | | ZsRE | | |
|---|---|---|---|---|---|---|---|---|---|
| | | Efficacy↑ | Generalization↑ | Specificity↑ | Fluency↑ | Consistency↑ | Efficacy↑ | Generalization↑ | Specificity↑ |
| | | | | Sequential Model Editing Task | | | | | |
| Pre-edited | Llama3 | $7.85_{\pm0.26}$ | $10.58_{\pm0.26}$ | $89.48_{\pm0.18}$ | $635.23_{\pm0.11}$ | $24.14_{\pm0.08}$ | $36.99_{\pm0.30}$ | $36.34_{\pm0.30}$ | $31.89_{\pm0.22}$ |
| PMET | | $99.47_{\pm0.26}$ | $90.78_{\pm0.84}$ | $76.07_{\pm0.92}$ | $619.62_{\pm0.66}$ | $32.45_{\pm0.41}$ | $94.97_{\pm0.45}$ | $89.98_{\pm0.75}$ | $32.95_{\pm0.81}$ |
| PMET$_{\text{BLUE}}$ | | $99.57_{\pm0.24}$ | $94.13_{\pm0.69}$ | $83.77_{\pm0.77}$ | $626.26_{\pm0.51}$ | $32.29_{\pm0.38}$ | $96.07_{\pm0.36}$ | $91.73_{\pm0.66}$ | $32.66_{\pm0.81}$ |
| Pre-edited | GPT-J | $16.22_{\pm0.31}$ | $18.56_{\pm0.45}$ | $83.11_{\pm0.13}$ | $621.81_{\pm0.67}$ | $29.74_{\pm0.51}$ | $26.32_{\pm0.37}$ | $25.79_{\pm0.25}$ | $27.42_{\pm0.53}$ |
| PMET | | $99.73_{\pm0.18}$ | $93.93_{\pm0.70}$ | $72.32_{\pm0.96}$ | $618.82_{\pm0.62}$ | $41.77_{\pm0.45}$ | $99.07_{\pm0.26}$ | $96.10_{\pm0.56}$ | $28.79_{\pm0.93}$ |
| PMET$_{\text{BLUE}}$ | | $99.57_{\pm0.24}$ | $92.82_{\pm0.76}$ | $77.61_{\pm0.92}$ | $620.02_{\pm0.59}$ | $39.19_{\pm0.43}$ | $99.16_{\pm0.25}$ | $87.37_{\pm1.01}$ | $28.13_{\pm0.93}$ |
| Pre-edited | GPT2-XL | $22.23_{\pm0.73}$ | $24.34_{\pm0.62}$ | $78.53_{\pm0.33}$ | $626.64_{\pm0.31}$ | $31.88_{\pm0.20}$ | $22.19_{\pm0.24}$ | $31.30_{\pm0.27}$ | $24.15_{\pm0.32}$ |
| PMET | | $95.80_{\pm0.72}$ | $87.27_{\pm1.00}$ | $62.66_{\pm1.06}$ | $542.47_{\pm2.49}$ | $31.56_{\pm0.54}$ | $93.22_{\pm0.69}$ | $87.06_{\pm0.96}$ | $25.58_{\pm0.91}$ |
| PMET$_{\text{BLUE}}$ | | $95.30_{\pm0.76}$ | $85.57_{\pm1.08}$ | $67.93_{\pm0.99}$ | $603.03_{\pm1.37}$ | $37.20_{\pm0.42}$ | $89.32_{\pm0.91}$ | $80.53_{\pm1.19}$ | $26.74_{\pm0.92}$ |
| | | | | Batch Model Editing Task | | | | | |
| Pre-edited | Llama3 | $7.02_{\pm0.50}$ | $9.44_{\pm0.49}$ | $89.73_{\pm0.36}$ | $630.00_{\pm0.22}$ | $24.21_{\pm0.17}$ | $35.67_{\pm0.58}$ | $34.81_{\pm0.58}$ | $31.83_{\pm0.44}$ |
| PMET | | $97.02_{\pm0.33}$ | $86.22_{\pm0.58}$ | $77.72_{\pm0.48}$ | $624.68_{\pm0.28}$ | $31.84_{\pm0.21}$ | $83.49_{\pm0.53}$ | $80.73_{\pm0.56}$ | $31.94_{\pm0.43}$ |
| PMET$_{\text{BLUE}}$ | | $93.64_{\pm0.48}$ | $81.52_{\pm0.67}$ | $84.63_{\pm0.40}$ | $627.81_{\pm0.24}$ | $30.62_{\pm0.20}$ | $85.92_{\pm0.50}$ | $82.83_{\pm0.54}$ | $32.23_{\pm0.44}$ |
| Pre-edited | GPT-J | $15.20_{\pm0.70}$ | $17.70_{\pm0.60}$ | $83.50_{\pm0.50}$ | $622.40_{\pm0.30}$ | $29.40_{\pm0.20}$ | $26.40_{\pm0.60}$ | $25.80_{\pm0.50}$ | $27.00_{\pm0.50}$ |
| PMET | | $99.57_{\pm0.13}$ | $92.48_{\pm0.44}$ | $71.41_{\pm0.52}$ | $620.31_{\pm0.31}$ | $40.79_{\pm0.24}$ | $89.24_{\pm0.46}$ | $82.69_{\pm0.59}$ | $25.51_{\pm0.49}$ |
| PMET$_{\text{BLUE}}$ | | $97.65_{\pm0.59}$ | $87.24_{\pm1.13}$ | $73.32_{\pm1.06}$ | $616.85_{\pm0.65}$ | $38.14_{\pm0.46}$ | $80.77_{\pm1.24}$ | $67.58_{\pm1.45}$ | $28.00_{\pm1.01}$ |
| Pre-edited | GPT2-XL | $21.82_{\pm0.81}$ | $24.16_{\pm0.72}$ | $78.32_{\pm0.55}$ | $626.78_{\pm0.23}$ | $31.37_{\pm0.20}$ | $22.17_{\pm0.52}$ | $21.28_{\pm0.51}$ | $24.20_{\pm0.48}$ |
| PMET | | $81.14_{\pm0.77}$ | $70.45_{\pm0.79}$ | $66.42_{\pm0.56}$ | $622.16_{\pm0.32}$ | $37.09_{\pm0.22}$ | $60.25_{\pm0.78}$ | $56.29_{\pm0.78}$ | $23.95_{\pm0.49}$ |
| PMET$_{\text{BLUE}}$ | | $67.09_{\pm0.92}$ | $54.48_{\pm0.87}$ | $73.48_{\pm0.54}$ | $626.67_{\pm0.25}$ | $35.05_{\pm0.21}$ | $57.15_{\pm0.77}$ | $51.99_{\pm0.77}$ | $25.01_{\pm0.48}$ |

# M    Impact Statements

The model editing studied in this paper aims to efficiently update outdated or incorrect knowledge in large language models (LLMs). While the original intention behind such techniques is beneficial, there is also potential for misuse, such as injecting false or malicious content into LLMs. Therefore, we caution readers not to place blind trust in the content generated by LLMs.

