# OpenReview forum: "Rethinking Residual Distribution in Locate-then-Edit Model Editing"
_NeurIPS.cc/2025/Conference — NeurIPS 2025 poster_

### Official Review · Reviewer_Z28o · 2025-06-29

**Clarity:** 3
**Significance:** 2
**Originality:** 2
**Rating:** 3
**Confidence:** 4

**Summary:**

This paper investigates a critical issue in locate-then-edit model editing methods for Large Language Models (LLMs): the negative impact of residual distribution on editing precision. Through both empirical and theoretical analyses, the authors demonstrate that residual distribution introduces weight shift errors that worsen with increased distribution distance, batch size, and edit sequence length, leading to suboptimal edits. To address this, they propose the Boundary Layer UpdatE (BLUE) strategy, which updates only the first and last critical layers by directly computing residuals, eschewing residual distribution. Experiments on three LLMs and two datasets show that BLUE significantly improves editing performance, better preserves LLM general capabilities, and mitigates hidden state shifts.

**Questions:**

none

**Ethical Concerns:**

["NO or VERY MINOR ethics concerns only"]

**Final Justification:**

refer to comments

**Limitations:**

yes

**Quality:**

2

**Strengths And Weaknesses:**

Strengths：
-  The paper effectively identifies and analyzes a counterintuitive failure mode in prominent locate-then-edit methods: residual distribution. The paper shows that evenly distributing residuals is suboptimal, leading to significant degradation in efficacy and generalization.
- A wide array of ablations and scenarios—including per-layer versus distributed residual comparison, sequential and standard batch editing, long-context editing, and general capability retention—validates BLUE’s superiority.

Weakness:
- The BLUE method proposed in this paper lacks novelty. As shown in Figure 1, the approach merely reallocates the residual updates from all layers to only the first and last layers. This strategy is overly simplistic and lacks significant innovation, offering limited insight or inspiration.
- The paper is overly lengthy and difficult to follow. In Section 4, the proofs regarding the upper and lower bounds of the interpolation error suggest that the upper bound of weight shift is related to the number of editing batches, the number of edits, and residual distribution. However, this does not demonstrate a strong correlation between the weight shift itself and these factors—let alone establish a strong connection to the proposed BLUE method.
- Many of the mathematical derivations and formulas in the paper are unclear, making it difficult to understand their meaning. For example, in Equation (2) in Section 3, the symbol n appears, but it is not clear what n represents. The surrounding text does not provide an explanation either. Additionally, the explanations of K₁ and M₁ are also confusing and hard to comprehend.

---

> ### Author Rebuttal · Authors · 2025-07-31
>
> We thank the reviewer for their efforts in reviewing our paper. We address the reviewer‘s concerns below:
>
> ---
>
> **W1: BLUE lacks novelty.**
>
>  We acknowledge that BLUE is a simple strategy since it only reallocates the residual updates from all layers to only the first and last layers. However, **BLUE is a highly effective method**. We demonstrate its effectiveness through experiments involving three models and two datasets in both sequential and batch editing settings. BLUE **boosts existing locate-then-edit methods**: it not only improves editing performance but also better preserves the model’s general capabilities and is more computationally efficient.
>  Additionally, through both empirical and theoretical analysis, we are the first to identify a counterintuitive phenomenon in current locate-then-edit model editing: **residual propagation can actually introduce weight shift errors**. Based on this insight, we propose the BLUE strategy to enhance locate-then-edit methods.
>
> ------
>
> **W2 (1): The paper is overly lengthy and difficult to follow.**
>
>  We acknowledge that the paper's organization could be improved, such as overly lengthy background and proof sections. We will optimize the structure of the paper in the next version.
>
> ------
>
>
> **W2 (2): Section 4 does not demonstrate a strong correlation between the weight shift itself and the factors of editing batches, the number of edits, and residual distribution.**
>
> We would like to clarify that our theorem and lemmas **characterize the weight shift error**—not the weight shift itself—that arises from factors such as editing batch size, number of edits, and residual distribution. Our theoretical results establish an upper bound on the weight shift error and link it to these factors. This connection forms the theoretical foundation for proposing the BLUE method. Instead of directly correlating weight shift with these factors, we **use the weight shift variable to bound the weight shift error**.
>
> ------
>
> **W3: Many of the mathematical derivations and formulas in the paper are unclear**.
>
>  We regret any confusion caused by insufficient explanations for some of the notations. In **Equation (2)**, $n$ denotes the number of memories to be updated. $K_1$ refers to the keys of the new memory, and $M_1$ denotes the corresponding values.  We will carefully review the entire paper and correct similar issues and include a more detailed description in the next version to make it easier to understand.

---

> > ### Comment · Reviewer_Z28o · 2025-08-03
> >
> > Thanks for the response and some concerns are clarified, and I will raise the Clarity score.

---

> > > ### Author Response · Authors · 2025-08-03
> > >
> > > Thank you for raising the Clarity score. If you have any further concerns, please feel free to let us know and we will be happy to provide additional responses.

---

> ### Author Response · Authors · 2025-08-05
> **A kindly reminder**
>
> We sincerely appreciate your valuable feedback and the constructive discussion so far. As the discussion phase is approaching its end (in around 30 hours), we just wanted to kindly check whether our responses have addressed your concerns. If there’s anything unclear or if you have further questions, we’d be more than happy to provide additional clarification.
>
> Thank you again for your time and efforts.

---

> > ### Author Response · Authors · 2025-08-06
> > **A kindly reminder**
> >
> > Dear Reviewer,
> >
> > Thank you again for your time and thoughtful feedback during the review process.
> >
> > As the discussion phase will conclude soon (on **August 8, 11:59pm AoE**), we would like to gently remind you that if you have any further questions or remaining concerns, we would be happy to address them before the deadline.
> >
> > Please don’t hesitate to reach out if there’s anything else we can clarify.

---

> ### Author Response · Authors · 2025-08-08
> **A Kindly Reminder – Discussion Phase Ending Soon**
>
> Dear Reviewer,
>
> Thank you again for your time and valuable comments during the review process.
>
> As the discussion phase will end in less than 24 hours (August 8, 11:59pm AoE), we would like to gently follow up to see if you have any further thoughts or feedback. If there are any remaining questions or concerns, we would be happy to clarify them before the deadline.
>
> We sincerely appreciate your contributions and engagement.
>
> Best regards

---

### Official Review · Reviewer_iak8 · 2025-06-30

**Clarity:** 3
**Significance:** 3
**Originality:** 2
**Rating:** 4
**Confidence:** 3

**Summary:**

This paper critically analyzes the residual distribution mechanism in the localize-then-edit editing methods for large language models (LLMs). The authors point out a key limitation - the residual distribution introduces weight-shift error, which degrades editing accuracy. To address this issue, they propose the BLUE strategy, which updates the first and last key layers using only directly computed residuals. Extensive experiments on multiple LLMs and datasets show that BLUE significantly improves editing performance while preserving the general capabilities of the model.

**Questions:**

This paper modifies two layer of models, but method like ROME only update one layer, why not only update one layer? This maybe the minimal residual distribution.

**Ethical Concerns:**

["NO or VERY MINOR ethics concerns only"]

**Final Justification:**

Some concerns have been addressed.

**Limitations:**

yes

**Quality:**

2

**Strengths And Weaknesses:**

Strengths:

1.	This paper reveals a intuitive and important problem in existing localize-then-edit methods - the residual distribution is not as effective as previously assumed. The authors provide theoretical upper bounds and empirical validation for the weight-shift error caused by the residual distribution. The analysis is rigorous and informative.

2.	The proposed BLUE strategy is simple, efficient, and widely applicable. It improves performance without significantly increasing computational cost and is easily integrated with multiple existing methods (MEMIT, PRUNE, RECT, AlphaEdit).

3.	It shows that BLUE consistently improves editing performance across different metrics and settings, including sequential batch editing, general ability preservation, and long-form editing.

Weaknesses

1.	While BLUE generally improves performance, in some cases (e.g., AlphaEdit on GPT-J), the improvement is small. A more detailed analysis of when and why BLUE may fail would help strengthen the robustness of this paper.

2.	This paper focus on models with parameter sizes up to 8B or 6B. It would be valuable to explore whether these findings generalize to larger state-of-the-art models.

3.	This paper focuses on least-squares-based editing methods. It is unclear how well BLUE generalizes to more complex or nonlinear optimization strategies, which may limit its applicability.

4. This paper modifies two layer of models, but method like ROME only update one layer, why not only update one layer? This maybe the minimal residual distribution.

5. This article has a bit of background content and may be simplified a bit.

---

> ### Author Rebuttal · Authors · 2025-07-31
>
> We thank the reviewer for acknowledging that the BLUE strategy is simple, efficient, and broadly applicable. We address the reviewer‘s concerns below:
>
> ---
>
> **W1: The improvement is small in some cases.**
>
> We acknowledge that in some cases (e.g., AlphaEdit on GPT-J), the improvements brought by BLUE are relatively small. However, in these cases, the baseline performance is already very high—close to 100—which makes achieving further gains particularly challenging. To gain a deeper understanding of the reasons behind BLUE's failure, we analyzed the **AlphaEdit on GPT-J** case in detail. Our investigation revealed that **BLUE performs less effectively when handling edits involving highly similar concepts**, such as the following two cases:
>
> - “What is the birthplace of Bas Verwijlen? Amsterdam → Oss”
> - “In which country did Bas Verwijlen live? Switzerland → Netherlands”
>
> In this example, **BLUE was only able to successfully edit the second case**, while **AlphaEdit succeeded in editing both**. This may be because **BLUE introduces stronger regularization**, which restricts the parameter expressiveness and causes the model to **forget the previously edited knowledge** when editing the latter case.
>
> We were able to fix this failure by using a **smaller $\alpha$** value. However, this leads to numerical instability when solving the equation, ultimately degrading overall performance. At present, we use a moderate $\alpha$ to ensure stable performance, and we plan to explore better solutions to address such failures in future work.
>
> Interestingly, we did not observe this issue in LLaMA3, which we attribute to architectural differences between LLaMA3 and GPT-J. We will include this failure case analysis in the final version of the paper.
>
> ---
>
> **W2: It would be valuable to explore whether these findings generalize to larger state-of-the-art models.**
>
> To address the reviewer’s concerns, we will conduct editing experiments on **LLaMA-2-13B**, and the results will be presented in the next few days.
>
> ---
>
> **W3: It is unclear how well BLUE generalizes to more complex or nonlinear optimization strategies.**
>
>  We acknowledge that BLUE is an optimization specifically designed for **least-squares-based editing methods**, and it may not yield similar benefits—or might not be applicable at all—for **more complex or nonlinear optimization strategies**.
>  However, we would like to emphasize that least-squares-based editing represents a **prominent paradigm in model editing**, under which many effective methods have been proposed, including the four methods we evaluated in this paper: **MEMIT, AlphaEdit, RECT, and PRUNE**.
>  Our results show that BLUE consistently improves performance across all of these methods. Moreover, least-squares-based editing has also been extended to more realistic **long-text editing scenarios**. As shown in **Table 4**, BLUE enhances performance in those settings as well.
>  Considering the broad adoption of least-squares-based editing methods, the potential application scope of BLUE is likewise extensive.
>
> ---
>
> **W4 & Q: The reason why we update two layers instead of one.**
>
>  From a **theoretical perspective**, locate-then-edit methods assume that each layer in the model encodes a portion of the complete knowledge and that the capacity of a single layer's weights is limited [1], making **single-layer updates insufficient** for large-scale knowledge modifications.
>  From an **empirical perspective**, **multi-layer updates** are a common practice in locate-then-edit methods. Prior work has demonstrated that multi-layer edits **outperform single-layer updates** [1,2].
>  Additionally, we conducted experiments on single-layer updates, and the results are as follows:
>
> ## CounterFact
>
> | Method    | Edit Layer | Efficacy | Generalization | Specificity | Fluency | Consistency |
> | --------- | ---------- | -------- | -------------- | ----------- | ------- | ----------- |
> | AlphaEdit | 8          |          |                |             |         |             |
> |           | 7          | 97.25    | 85.28          | 76.75       | 559.78  | 4.01        |
> |           | 6          | 96.25    | 85.02          | 79.16       | 560.79  | 3.53        |
> |           | 5          | 94.3     | 86.92          | 81.68       | 561.12  | 4.03        |
> |           | 4          | 92.75    | 83.15          | 83.16       | 556.41  | 3.66        |
> |           | BLUE       | 99.93    | 97.25          | 75.24       | 624.90  | 33.79       |
> | MEMIT     | 8          | 94.05    | 73.42          | 72.30       | 556.47  | 3.6         |
> |           | 7          | 95.15    | 81.38          | 71.67       | 560.09  | 3.84        |
> |           | 6          | 96.4     | 82.8           | 75.09       | 563.42  | 3.96        |
> |           | 5          | 97.8     | 90.98          | 79.12       | 563.29  | 4.21        |
> |           | 4          | 97.8     | 93.2           | 81.54       | 555.66  | 3.72        |
> |           | BLUE       | 99.57    | 94.13          | 83.77       | 626.66  | 32.29       |
>
> ## zsre
>
> | Method    | Edit Layer | Efficacy | Generalization | Specificity |
> | --------- | ---------- | -------- | -------------- | ----------- |
> | AlphaEdit | 8          | 92.97    | 89.46          | 30.68       |
> |           | 7          | 92.79    | 90.88          | 30.91       |
> |           | 6          | 92.75    | 86.50          | 33.10       |
> |           | 5          | 91.21    | 84.05          | 33.07       |
> |           | 4          | 86.42    | 80.92          | 32.83       |
> |           | BLUE       | 95.77    | 91.73          | 31.96       |
> | MEMIT     | 8          | 91.96    | 86.12          | 32.98       |
> |           | 7          | 95.54    | 87.09          | 33.37       |
> |           | 6          | 92.76    | 86.62          | 33.33       |
> |           | 5          | 92.22    | 85.27          | 33.29       |
> |           | 4          | 92.95    | 87.98          | 33.31       |
> |           | BLUE       | 95.94    | 90.98          | 32.41       |
>
> The results above show that single-layer updates perform significantly worse than BLUE’s two-layer update on both datasets, indicating that the two-layer update achieves better editing performance.
>
> ---
>
> **W5: This article has a bit of background content and may be simplified a bit.**
>
> In the next version, we will simplify the background section of the paper and place **more core empirical evidence supporting BLUE’s design** in the main text.
>
> ---
>  [1] *Mass-Editing Memory in a Transformer*, ICLR 2023
>
>  [2] *AlphaEdit: Null-Space Constrained Knowledge Editing for Language Models*, ICLR 2025

---

> > ### Author Response · Authors · 2025-08-02
> > **Supplementary results for LLaMA-2-13B and W4 & Q**
> >
> > # Supplementary results for LLaMA-2-13B
> >
> > We selected AlphaEdit and MEMIT to perform sequential editing experiments on layers 30–34 of LLaMA-2-13B, following the same setup as in the main experiments. The results are shown in the table below.
> >
> > ### **CounterFact Dataset**
> >
> > | Method             | Efficacy               | Generalization         | Specificity        | Fluency                 | Consistency            |
> > | ------------------ | ---------------------- | ---------------------- | ------------------ | ----------------------- | ---------------------- |
> > | AlphaEdit          | 52.30$_{\pm 2.19}$     | 48.95$_{\pm 1.73}$     | 50.53$_{\pm 2.21}$ | 408.92$_{\pm 2.36}$     | 0.22$_{\pm 0.01}$      |
> > | AlphaEdit$_{BLUE}$ | **78.75**$_{\pm 1.79}$ | **69.53**$_{\pm 1.81}$ | 44.27$_{\pm 1.76}$ | **513.49**$_{\pm 3.09}$ | **22.10**$_{\pm 0.57}$ |
> > | MEMIT              | 79.20$_{\pm 1.77}$     | 65.94$_{\pm 1.79}$     | 41.53$_{\pm 1.75}$ | 378.10$_{\pm 4.01}$     | 14.44$_{\pm 0.59}$     |
> > | MEMIT$_{BLUE}$     | **79.60**$_{\pm 1.78}$ | **66.00**$_{\pm 1.81}$ | 41.28$_{\pm 1.65}$ | **384.23**$_{\pm 3.50}$ | **14.65**$_{\pm 0.59}$ |
> >
> > ------
> >
> > ### **ZsRE Dataset**
> >
> > | Method             | Efficacy               | Generalization         | Specificity           |
> > | ------------------ | ---------------------- | ---------------------- | --------------------- |
> > | AlphaEdit          | 52.31$_{\pm 1.44}$     | 37.68$_{\pm 1.49}$     | 9.11$_{\pm 0.60}$     |
> > | AlphaEdit$_{BLUE}$ | **53.01**$_{\pm 2.32}$ | **41.32**$_{\pm 1.52}$ | **9.18**$_{\pm 0.60}$ |
> > | MEMIT              | 50.11$_{\pm 1.46}$     | 34.70$_{\pm 1.47}$     | 9.71$_{\pm 0.61}$     |
> > | MEMIT$_{BLUE}$     | **51.07**$_{\pm 1.45}$ | **37.13**$_{\pm 1.52}$ | 9.07$_{\pm 0.60}$     |
> >
> > The results demonstrate that the **BLUE-enhanced methods** achieve **better overall editing performance** than the baseline methods on both the CounterFact and ZsRE datasets. We bolded all the enhanced results. On the **CounterFact** dataset, all metrics except for Specificity show improvements over the baseline. On the **ZsRE** dataset, all metrics of AlphaEdit show improvements, while MEMIT exhibits improvements in all metrics except for Specificity. This indicates that BLUE is also **effective on LLaMA-2-13B**, further validating the effectiveness of our approach.
> >
> > ---
> >
> > # Supplementary results for W4 & Q
> >
> > We apologize for the oversight in our previous response, where we omitted the results of AlphaEdit editing only the 8th layer mentioned in our reply to **W4 & Q**. These results had been obtained earlier but were inadvertently left out of the table in the final submission. We now provide the complete results as follows:
> >
> > | Method    | Edit Layer | Efficacy | Generalization | Specificity | Fluency | Consistency |
> > | --------- | ---------- | -------- | -------------- | ----------- | ------- | ----------- |
> > | AlphaEdit | 8          | 96.50    | 80.95          | 76.24       | 556.53  | 3.54        |
> > |           | 7          | 97.25    | 85.28          | 76.75       | 559.78  | 4.01        |
> > |           | 6          | 96.25    | 85.02          | 79.16       | 560.79  | 3.53        |
> > |           | 5          | 94.3     | 86.92          | 81.68       | 561.12  | 4.03        |
> > |           | 4          | 92.75    | 83.15          | 83.16       | 556.41  | 3.66        |
> > |           | BLUE       | 99.93    | 97.25          | 75.24       | 624.90  | 33.79       |
> > | MEMIT     | 8          | 94.05    | 73.42          | 72.30       | 556.47  | 3.6         |
> > |           | 7          | 95.15    | 81.38          | 71.67       | 560.09  | 3.84        |
> > |           | 6          | 96.4     | 82.8           | 75.09       | 563.42  | 3.96        |
> > |           | 5          | 97.8     | 90.98          | 79.12       | 563.29  | 4.21        |
> > |           | 4          | 97.8     | 93.2           | 81.54       | 555.66  | 3.72        |
> > |           | BLUE       | 99.57    | 94.13          | 83.77       | 626.66  | 32.29       |
> >
> > We apologize once again for our oversight.

---

> > ### Comment · Reviewer_iak8 · 2025-08-06
> >
> > Thanks for your response. I have raised my score.

---

> > > ### Author Response · Authors · 2025-08-06
> > >
> > > Thank you very much for your kind response and for your willingness to raise the score. We truly appreciate your time and constructive feedback throughout the review process. Your comments have been very helpful in improving our work.

---

> ### Author Response · Authors · 2025-08-05
> **A kindly reminder**
>
> We sincerely appreciate your valuable and constructive feedback. As the discussion phase is approaching its end (in around 30 hours), we just wanted to kindly check whether our responses have addressed your concerns. If there’s anything unclear or if you have further questions, we’d be more than happy to provide additional clarification.
>
> Thank you again for your time and efforts.

---

> > ### Author Response · Authors · 2025-08-06
> > **A kindly reminder**
> >
> > Dear Reviewer,
> >
> > Thank you again for your time and thoughtful feedback during the review process.
> >
> > As the discussion phase will conclude soon (on **August 8, 11:59pm AoE**), we would like to gently remind you that if you have any further questions or remaining concerns, we would be happy to address them before the deadline.
> >
> > Please don’t hesitate to reach out if there’s anything else we can clarify.

---

### Official Review · Reviewer_bKjA · 2025-07-02

**Clarity:** 3
**Significance:** 3
**Originality:** 3
**Rating:** 4
**Confidence:** 1

**Summary:**

This paper studies model editing for large language models (LLMs), focusing on the prevalent locate-then-edit paradigm. The authors identify that the residual distribution mechanism used in existing methods introduces weight shift errors, which worsen with increased distribution distance, batch size, and edit sequence length, reducing editing accuracy. To address this, they propose the Boundary Layer UpdatE (BLUE) strategy to mitigate these errors. Experiments on three LLMs and two datasets demonstrate that BLUE improves editing performance by an average of 35.59% while better preserving the general capabilities of LLMs, significantly advancing the state of the art in model editing.

**Questions:**

1. The BLUE method updates only the first and last critical layers without residual distribution. How sensitive is the method’s performance to the selection of these critical layers? Would different choices affect editing accuracy or model preservation?
2. The experiments are conducted on three LLMs and two datasets. Could the authors clarify whether BLUE generalizes well to larger-scale models (e.g., GPT-3 scale) or other architectures beyond those tested?
3. Does the BLUE strategy introduce additional computational or memory overhead compared to existing locate-then-edit methods? If so, how significant is it in practical scenarios?

**Ethical Concerns:**

["NO or VERY MINOR ethics concerns only"]

**Limitations:**

yes

**Quality:**

3

**Strengths And Weaknesses:**

Strengths:

1. The paper is well-written and easy to follow.
2. The proposed method is well-motivated.
3. The experiment is thorough and convincing.

Weakness:
1. The authors do not explicitly discuss the computational overhead or efficiency trade-offs introduced by BLUE compared to standard locate-then-edit methods, which may be important for practical deployment.
2. While the paper mentions improvements in long-form editing scenarios, more detailed analysis or examples of such use cases would strengthen understanding of BLUE’s benefits in real-world editing tasks.

---

> ### Author Rebuttal · Authors · 2025-07-31
>
> We thank the reviewer for recognizing that **our paper is well-written and easy to follow**, that **the proposed method is well-motivated**, and that **the experiment is thorough and convincing**. We will address the reviewer’s concerns one by one.
>
>
>
> ---
>
> **W1 & Q3: Whether the BLUE strategy introduces any additional computational or memory overhead compared to existing locate-then-edit methods.**
>
> We appreciate the reviewer’s detailed comments. We would like to discuss the efficiency of BLUE from both **time** and **memory** perspectives.
>
> In terms of **time efficiency**, BLUE improves the efficiency of locate-then-edit methods. Although BLUE requires computing residuals twice, it only updates two key layers of the original locate-then-edit method. This reduces the computational overhead and enhances time efficiency. We provide an analysis of BLUE's time efficiency in **Appendix I (Efficiency Analysis)**. The results show that BLUE indeed improves the time efficiency of locate-then-edit methods.
>
> In terms of **memory efficiency**, we report the **peak memory usage** of both BLUE and locate-then-edit in the table below:
>
> |                    | CounterFact |       |         | ZsRE    |       |         |
> | ------------------ | ----------- | ----- | ------- | ------- | ----- | ------- |
> |                    | Llama-3     | GPT-J | GPT-2XL | Llama-3 | GPT-J | GPT-2XL |
> | AlphaEdit          | 36.09       | 30.43 | 7.14    | 36.09   | 30.43 | 7.43    |
> | AlphaEdit$_{BLUE}$ | 35.10       | 28.92 | 7.24    | 35.43   | 28.92 | 7.49    |
> | MEMIT              | 37.07       | 31.68 | 7.34    | 37.07   | 31.68 | 7.34    |
> | MEMIT$_{BLUE}$     | 36.42       | 30.67 | 7.30    | 36.42   | 30.67 | 7.38    |
> | RECT               | 37.07       | 31.68 | 7.34    | 37.07   | 31.68 | 7.34    |
> | RECT$_{BLUE}$      | 36.42       | 30.67 | 7.22    | 36.42   | 30.67 | 7.37    |
> | PRUNE              | 38.17       | 33.18 | 7.53    | 38.17   | 33.18 | 7.53    |
> | PRUNE$_{BLUE}$     | 36.85       | 30.67 | 7.30    | 36.85   | 30.67 | 7.46    |
>
> The results show that BLUE achieves **better memory efficiency** compared to the original method. This outcome is intuitive: by discarding residual distribution and instead computing residuals only for the first and last key layers, BLUE reduces the memory usage associated with the residual distribution process.
>
>
>
> ---
>
> **W2: More detailed analysis and cases in long-form editing scenarios should be included.**
>
> We thank the reviewer for the constructive suggestion. We will include **case studies** to provide a more detailed analysis of **BLUE’s benefits in real-world editing tasks**. For the reviewer’s convenience, we analyze two cases as follows:
>
> | Type               | Text                                                         |
> | ------------------ | ------------------------------------------------------------ |
> | Input              | What is George Rankin's occupation?                          |
> | Paraphrase         | What does George Rankin do for a living?                     |
> | Target Output      | George Rankin has been actively involved in politics for over a decade. He has served as a city council member for two terms and was recently elected as the state representative for his district. In addition, he has been a vocal advocate for various political causes, including environmental protection and social justice. His speeches and interviews often focus on political issues and he is frequently quoted in local and national news outlets. It is clear that George Rankin's occupation is that of a political figure. |
> | AlphaEdit          | George Rankin has been involved in politics for over 20 years. He has served as a member of the state legislature for 10 years and has been a mayor for 5 years. He has also been a member of various political committees and has run for several political offices, including governor and U.S. Senate. |
> | MEMIT              | George Rankin has been involved in politics for over 20 years. He has served as a member of the state legislature for 10 years and has been a mayor for 5 years. He has also run for governor and senator, but has not held those offices. |
> | AlphaEdit$_{BLUE}$ | George Rankin has been actively involved in politics for over a decade. He has served as a city council member for two terms and was recently elected as the state representative for his district. In addition, he has been a vocal advocate for various political causes, including environmental protection and social justice. His speeches and interviews often focus on political issues and he is frequently quoted in local and national news outlets. It is clear that George Rankin has been involved in politics for a long time and continues to be an active and influential figure in the political scene. |
> | MEMIT$_{BLUE}$     | George Rankin has been actively involved in politics for over a decade. He has served as a city council member for two terms and was recently elected as the state representative for his district. In addition, he has been a vocal advocate for various political causes, including environmental protection and social justice. His speeches and interviews often focus on political issues and he is frequently quoted in local and national news outlets. It is clear that George Rankin's involvement in politics is a significant aspect of his public persona and identity. |
>
> It can be seen that the method enhanced by BLUE generates content that is **noticeably closer** to the target output compared to the original method.
>
> ---
> > **Q1: The BLUE method updates only the first and last critical layers without residual distribution. How sensitive is the method’s performance to the selection of these critical layers? Would different choices affect editing accuracy or model preservation?**
>
> In **Appendix H**, we conducted an ablation study on the **Llama3 model** to demonstrate that the selection of key layers indeed affects both editing accuracy and model preservation.
>
> For $AlphaEdit_{BLUE}$ and MEMIT$_{BLUE}$, different layer selections slightly impact editing accuracy, but have a larger effect on generalization and specificity (i.e., model preservation). In particular, choosing layers 7 and 8 leads to the most significant drop in generalization and specificity.
>
> For $RECT_{BLUE}$ and PRUNE$_{BLUE}$, different layer choices significantly affect both **editing accuracy** and **model preservation**. For example, choosing layers 6 and 7 causes a sharp decline in RECT’s editing performance, while selecting layers 7 and 8 greatly harms PRUNE’s performance.
>
> Furthermore, the results of the ablation study suggest that choosing the **first and last key layers** as editing layers, as in BLUE, yields the best editing performance.
>
>
>
> ---
>
> >**Q2: The experiments are conducted on three LLMs and two datasets. Could the authors clarify whether BLUE generalizes well to larger-scale models (e.g., GPT-3 scale) or other architectures beyond those tested?**
>
> Due to limited GPU resources, we are currently unable to scale the editing experiments to **GPT-3-scale LLMs**. To address this concern raised by the reviewer, we will conduct editing experiments on **LLaMA-2-13B**, and the results will be presented in the next few days.

---

> > ### Author Response · Authors · 2025-08-02
> > **Supplementary results for LLaMA-2-13B**
> >
> > We selected AlphaEdit and MEMIT to perform sequential editing experiments on layers 30–34 of LLaMA-2-13B, following the same setup as in the main experiments. The results are shown in the table below.
> >
> > ### **CounterFact Dataset**
> >
> > | Method             | Efficacy               | Generalization         | Specificity        | Fluency                 | Consistency            |
> > | ------------------ | ---------------------- | ---------------------- | ------------------ | ----------------------- | ---------------------- |
> > | AlphaEdit          | 52.30$_{\pm 2.19}$     | 48.95$_{\pm 1.73}$     | 50.53$_{\pm 2.21}$ | 408.92$_{\pm 2.36}$     | 0.22$_{\pm 0.01}$      |
> > | AlphaEdit$_{BLUE}$ | **78.75**$_{\pm 1.79}$ | **69.53**$_{\pm 1.81}$ | 44.27$_{\pm 1.76}$ | **513.49**$_{\pm 3.09}$     | **22.10**$_{\pm 0.57}$ |
> > | MEMIT              | 79.20$_{\pm 1.77}$     | 65.94$_{\pm 1.79}$     | 41.53$_{\pm 1.75}$ | 378.10$_{\pm 4.01}$     | 14.44$_{\pm 0.59}$     |
> > | MEMIT$_{BLUE}$     | **79.60**$_{\pm 1.78}$ | **66.00**$_{\pm 1.81}$ | 41.28$_{\pm 1.65}$ | **384.23**$_{\pm 3.50}$ | **14.65**$_{\pm 0.59}$     |
> >
> > ------
> >
> > ### **ZsRE Dataset**
> >
> > | Method             | Efficacy               | Generalization         | Specificity           |
> > | ------------------ | ---------------------- | ---------------------- | --------------------- |
> > | AlphaEdit          | 52.31$_{\pm 1.44}$     | 37.68$_{\pm 1.49}$     | 9.11$_{\pm 0.60}$     |
> > | AlphaEdit$_{BLUE}$ | **53.01**$_{\pm 2.32}$ | **41.32**$_{\pm 1.52}$ | **9.18**$_{\pm 0.60}$ |
> > | MEMIT              | 50.11$_{\pm 1.46}$     | 34.70$_{\pm 1.47}$     | 9.71$_{\pm 0.61}$     |
> > | MEMIT$_{BLUE}$     | **51.07**$_{\pm 1.45}$ | **37.13**$_{\pm 1.52}$ | 9.07$_{\pm 0.60}$     |
> >
> > The results demonstrate that the **BLUE-enhanced methods** achieve **better overall editing performance** than the baseline methods on both the CounterFact and ZsRE datasets. We bolded all the enhanced results. On the **CounterFact** dataset, all metrics except for Specificity show improvements over the baseline. On the **ZsRE** dataset, all metrics of AlphaEdit show improvements, while MEMIT exhibits improvements in all metrics except for Specificity. This indicates that BLUE is also **effective on LLaMA-2-13B**, further validating the effectiveness of our approach.

---

> ### Author Response · Authors · 2025-08-05
> **A kindly reminder**
>
> We sincerely appreciate your valuable and constructive feedback. As the discussion phase is approaching its end (in around 30 hours), we just wanted to kindly check whether our responses have addressed your concerns. If there’s anything unclear or if you have further questions, we’d be more than happy to provide additional clarification.
>
> Thank you again for your time and efforts.

---

> > ### Comment · Reviewer_bKjA · 2025-08-05
> > **Official Comment by Reviewer bKjA**
> >
> > Thanks for your response. I keep my score.

---

> > > ### Author Response · Authors · 2025-08-05
> > >
> > > Thank you for your response and for maintaining your score. We noticed that the confidence score is relatively low (1). If there are any aspects of the paper or our rebuttal that remain unclear or unconvincing, we would greatly appreciate your feedback. We are more than happy to provide further clarifications if needed. We truly value your feedback and would like to ensure that all of your concerns are fully addressed.

---

> > > > ### Author Response · Authors · 2025-08-06
> > > > **A kindly reminder**
> > > >
> > > > Dear Reviewer,
> > > >
> > > > Thank you again for your time and thoughtful feedback during the review process.
> > > >
> > > > As the discussion phase will conclude soon (on **August 8, 11:59pm AoE**), we would like to gently remind you that if you have any further questions or remaining concerns, we would be happy to address them before the deadline.
> > > >
> > > > Please don’t hesitate to reach out if there’s anything else we can clarify.

---

> ### Author Response · Authors · 2025-08-08
> **A Kindly Reminder – Discussion Phase Ending Soon**
>
> Dear Reviewer,
>
> Thank you again for your time and valuable comments during the review process.
>
> As the discussion phase will end in less than 24 hours (August 8, 11:59pm AoE), we would like to gently follow up to see if you have any further thoughts or feedback. If there are any remaining questions or concerns, we would be happy to clarify them before the deadline.
>
> We sincerely appreciate your contributions and engagement.
>
> Best regards

---

### Official Review · Reviewer_meft · 2025-07-02

**Clarity:** 3
**Significance:** 3
**Originality:** 3
**Rating:** 5
**Confidence:** 4

**Summary:**

This paper presents a critical analysis of the 'locate-then-edit' paradigm in model editing, specifically targeting the widely-used residual distribution mechanism. The authors compellingly argue, through both theoretical and empirical evidence, that this distribution method is a source of significant error that harms editing precision. To address this, they propose a simple yet effective strategy, Boundary Layer UpdatE (BLUE), which modifies only the first and last critical layers with directly computed residuals, bypassing the distribution step entirely. The authors validate their approach through extensive experiments on multiple LLMs and datasets, showing that BLUE not only improves editing performance but also better preserves the model's general capabilities and is more computationally efficient.

**Questions:**

None

**Ethical Concerns:**

["NO or VERY MINOR ethics concerns only"]

**Final Justification:**

I have checked all the rebuttals between reviewers and authors; I think the author has addressed the concern of the elaboration.
The only issue left is the multi-hop reasoning evaluation; the author has acknowledged it in the limitation.
I think this would make the paper better if the author would add this.

**Limitations:**

The author acknowledges the limitations in Appendix L and shows that the methods didn't evaluate the results in multi-hop reasoning benchmarks like MQuAKE.

**Paper Formatting Concerns:**

The format of the paper is proper.

**Quality:**

3

**Strengths And Weaknesses:**

### Strength
- The author conducts sufficient empirical analysis and theoretical analysis about the shortcomings of residual distribution. The empirical analysis in Section 4.1, particularly the experiments showing the diminishing contribution of distributed residuals (Figure 2) and their sub-optimality compared to directly computed ones (Figure 4), is clear and convincing. This empirical work is well-supported by a theoretical analysis (Theorem 4.1 and Lemma 4.3) that provides an upper bound on the weight shift error, formalizing the intuition that errors increase with distribution distance, batch size, and the number of sequential edits.
- The authors apply BLUE to multiple strong baselines (MEMIT, AlphaEdit, PRUNE, RECT) across several LLMs (Llama3, GPT-J, GPT2-XL) and standard datasets (CounterFact, zsRE). The results in Table 2 strongly support the primary claims of improved editing performance in sequential batch scenarios. Furthermore, the analysis of general capability retention on downstream GLUE tasks (Figure 6) and the mitigation of hidden state shifts (Figure 7) are particularly compelling, as they address critical aspects of model editing beyond simple efficacy and demonstrate a clear advantage for BLUE in preserving the integrity of the original model.


### Weakness
- While the justification for updating the first critical layer is well-motivated (as it is most affected by distribution error), the motivation for choosing the last critical layer is stated simply as "follow[ing] prior work" (Section 4.3, page 7). The ablation study in Appendix H (Figure 11) provides strong empirical validation for this choice, demonstrating its optimality. However, the main text would benefit from a more explicit theoretical or intuitive justification for why the first and last layers form the optimal pair, rather than, for instance, the first two critical layers.
- I think the proof section is a bit verbose in the paper, and I recommend the author move some of them to the appendix and show more experiements in the main text. To enhance the paper's narrative impact, I would recommend condensing this section slightly. The reclaimed space could be used to feature a crucial experimental result that is currently in the appendix. For example, the layer-selection ablation study (Appendix H, Figure 11) provides the core empirical evidence for BLUE's design. Elevating this experiment to the main paper would proactively answer a key question about the "Boundary Layer" choice and make the justification for the method even more compelling.
- The authors correctly identify that the method's performance on multi-hop reasoning questions is a limitation and an area for future work. This points to a more significant consideration regarding the scope of the paper's claims. The current evaluation focuses on a standard but specific type of generalization: applying a factual edit to paraphrased prompts. While the results are strong in this domain, the paper would be more impactful if it contextualized its findings more explicitly in the main text.

---

> ### Author Rebuttal · Authors · 2025-07-31
>
> We thank the reviewer for recognizing that **BLUE is a simple yet effective strategy**. We address the reviewer's concerns below:
>
> ---
> **W1: The main text needs to elaborate more on why the second layer is chosen as the last key layer rather than the second layer.**
>
>  We appreciate the reviewer’s detailed comment. On the one hand, BLUE is an optimization strategy, and we aim to preserve the mechanism used in existing locate-then-edit model editing methods, which compute residuals at the last key layer, to ensure broader applicability. On the other hand, the choice of the first key layer is based on this mechanism: when residuals are computed at the last key layer, the layer most affected by residual distribution is the first key layer. Therefore, we choose the last key layer as the second update layer. We will add this explanation to the main text.
>
> ---
> **W2: The proof section is a bit verbose.**
>
>  We thank the reviewer for the constructive suggestion. We will condense the proof section (Section 4), move *Remark 4.2* and *Lemma 4.3* to the Appendix, and integrate the content of *Appendix H* into the main text.
>
> ---
> **W3: Contextualizing the limitation about multi-hop reasoning more explicitly in the main text.**
>
>  We thank the reviewer for the helpful feedback. We will explicitly discuss the method’s limitation on multi-hop reasoning tasks and mention it as a direction for future work in the conclusion of the main text.

---

> ### Author Response · Authors · 2025-08-05
> **A kindly reminder**
>
> We sincerely appreciate your valuable and constructive feedback. As the discussion phase is approaching its end (in around 30 hours), we just wanted to kindly check whether our responses have addressed your concerns. If there’s anything unclear or if you have further questions, we’d be more than happy to provide additional clarification.
>
> Thank you again for your time and efforts.

---

> > ### Author Response · Authors · 2025-08-06
> > **A kindly reminder**
> >
> > Dear Reviewer,
> >
> > Thank you again for your time and thoughtful feedback during the review process.
> >
> > As the discussion phase will conclude soon (on **August 8, 11:59pm AoE**), we would like to gently remind you that if you have any further questions or remaining concerns, we would be happy to address them before the deadline.
> >
> > Please don’t hesitate to reach out if there’s anything else we can clarify.

---

> ### Author Response · Authors · 2025-08-08
> **A Kindly Reminder – Discussion Phase Ending Soon**
>
> Dear Reviewer,
>
> Thank you again for your time and for reviewing our paper.
>
> As the discussion phase will end in less than **24 hours** (August 8, 11:59pm AoE), we wanted to gently follow up. We noticed that you have not yet responded to our rebuttal, and we would greatly appreciate any feedback you might be able to share before the deadline. If there are any remaining questions or concerns, we’d be happy to clarify them.
>
> We sincerely appreciate your time and engagement.
>
> Best regards

---

> > ### Comment · Reviewer_meft · 2025-08-08
> >
> > Thanks for your reply; I have changed the score accordingly.

---

> > > ### Author Response · Authors · 2025-08-09
> > >
> > > Thank you very much for your response. We truly appreciate your time and constructive feedback throughout the review process, which have been very helpful in improving our paper.

---

### Decision · Program_Chairs · 2025-09-17

**Decision:**

Accept (poster)

**Comment:**

This paper provides a thorough and convincing analysis of the shortcomings of residual distribution in locate-then-edit methods. The combination of strong empirical evidence  and solid theoretical analysis makes the case compelling. The proposed BLUE method is simple, efficient, and broadly applicable, showing consistent improvements across multiple editing baselines (MEMIT, PRUNE, RECT, AlphaEdit), models (Llama3, GPT-J, GPT2-XL), and datasets (CounterFact, zsRE).

The experiments are extensive and well-designed, covering sequential batch edits, general capability retention (GLUE), and long-form editing, all of which strengthen the claims. BLUE improves editing efficacy without significant overhead and integrates smoothly with existing methods.

Overall, the paper is well-written and addresses an important and underexplored failure mode in model editing. Most reviewers agree this paper is above the acceptance borderline and recommend acceptance.